# Existence and influence of mixed states in a model of vegetation patterns

Lilian Vanderveken[1], Marina  Martínez Montero[1], and Michel Crucifix[1]

[1]Earth and Life Institute, Louvain-la-neuve, Belgium

**Correspondence:** Lilian Vanderveken (lilian.vanderveken@uclouvain.be)

**Abstract.** The Rietkerk vegetation model is a system of partial differential equations, which has been used to understand the formation and dynamics of spatial patterns in vegetation ecosystems, including desertification and biodiversity loss. Here, we provide an in-depth bifurcation analysis of the vegetation patterns produced by Rietkerk's model, based on a linear stability analysis of the homogeneous equilibrium of the system. Specifically, using a continuation method based on the Newton-Raphson algorithm, we obtain all the main heterogeneous equilibria for a given size of the domain. We confirm that inhomogeneous vegetated states can exist and be stable, even for a value of rainfall for which no vegetation exists in the non-spatialized system. In addition, we evidence the existence of a new type of equilibrium, which we called "mixed state", in which the equilibria are always unstable and take the form of a mix of two equilibria from the main branches. Although these equilibria are unstable, they influence the dynamics of the transitions between distinct stable states, by slowing down the evolution of the system when it passes close to it. Our approach proves to be a helpful way to assess the existence of tipping points in spatially extended systems and disentangle the fate of the system in the Busse balloon. Overall, our findings represent a significant step forward in understanding the behavior of the Rietkerk model and the broader dynamics of vegetation patterns.

## 1   Introduction

In semi-arid regions, vegetation tends to be spatially organised around patterns (Barbier et al. (2006), Deblauwe et al. (2008), Deblauwe et al. (2011), Deblauwe et al. (2012)). This phenomenon appears in various parts of the world where water is the limiting factor for plants' growth. Vegetation patterns can be modelled and explained with reaction-diffusion equations. Those types of equation exhibit the existence of a homogneous stable equilibrium which is unstable to heterogeneous perturbation (Turing, 1952). In general terms, heterogeneous equilibria result from the joint effects of a short-range activation mechanism and long-range inhibition. For vegetation on ferruginous soil in semi-arid regions, the short-range activation effect is related to the positive feedback of vegetation on soil water availability, as vegetation limits water loss by runoff by enhancing water infiltration (Meron, 2015).

The rapid diffusion of surface water, however, acts against vegetation growth on bare soil by limiting water availability. The contrast between slow soil water diffusion and fast surface water diffusion produces the spatial, heterogeneous patterns. The mechanisms are described and captured in the Rietkerk model (Rietkerk et al., 2002). This model features three (prognostic) variables: biomass ($B$) [g.m$^{-2}$], soil water ($W$) [mm] and surface water ($O$) [mm], all are functions of time and space. The

| $c$ | Conversion of water uptake by plants to plant growth | $10\,\mathrm{g\,mm^{-1}\,m^{-2}}$ |
|---|---|---|
| $g_{max}$ | Maximum water uptake | $0.05\,\mathrm{mm\,g^{-1}\,m^{-2}\,d^{-1}}$ |
| $k_1$ | Half-saturation constant of specific plant growth and water uptake | $5\,\mathrm{mm}$ |
| $D_B$ | Plant dispersal | $0.1\,\mathrm{m^2\,d^{-1}}$ |
| $\alpha$ | Maximum infiltration rate | $0.2\,\mathrm{d^{-1}}$ |
| $k_2$ | Saturation constant of water infiltration | $5\,\mathrm{g\,m^{-2}}$ |
| $w_0$ | Water infiltration in the absence of plants | $0.2$ |
| $r_w$ | Soil water loss due to evaporation and drainage | $0.2\,\mathrm{d^{-1}}$ |
| $D_W$ | Diffusion coefficient for soil water | $0.1\,\mathrm{m^2\,d^{-1}}$ |
| $D_O$ | Diffusion coefficient for surface water | $100\,\mathrm{m^2\,d^{-1}}$ |
| $d$ | Plant mortality rate | $0.25\,\mathrm{d^{-1}}$ |

**Table 1.** Parameters for Rietkerk's model

evolution of those quantities are governed by three equations:

$$
\begin{aligned}
\frac{\partial B}{\partial t} &= c g_{max} \frac{W B}{W + k_1} - dB + D_B \Delta B, \\
\frac{\partial W}{\partial t} &= \alpha O \frac{B + k_2 w_0}{B + k_2} - g_{max} \frac{W B}{W + k_1} - r_w W + D_W \Delta W, \\
\frac{\partial O}{\partial t} &= R - \alpha O \frac{B + k_2 w_0}{B + k_2} + D_O \Delta O,
\end{aligned}
\tag{1}
$$

where $\Delta$ is the Laplacian operator and $R$ is the rainfall [mm.d$^{-1}$]. The rainfall is the external forcing of the system which we consider to be a spatially-independent function. The first term in the biomass equation represents water uptake by the plant. The first term in the soil water equation is linked to the infiltration rate of water in the soil that is enhanced by the presence of biomass. The factors in front of the Laplacians ($\Delta B$, $\Delta W$ and $\Delta O$) are the diffusion constants of the different quantities. The diffusion constant for surface water is considered to be much bigger than those of biomass and soil water. This contrast is essential for pattern creation. In the following, the values of the parameters are as in Rietkerk et al. (2002).

As explained above, the existence of reaction-diffusion processes enables stable equilibria in the form of patterns. Compared to a system without spatial dynamics, such pattern equilibria tend to broaden the range of rainfall compatible with the presence of vegetation.

The classical configuration for analysing such a system of equations uses periodic boundary conditions. A pattern is then defined as a spatially periodic equilibrium of the differential equations. Here, we will consider and analyse in depth a region of the parameter space called the Busse balloon (Busse, 1978). This is the region with heterogeneous equilibria, that is, the parameter space admitting at least one stable, spatially periodic equilibrium.

The motivating scientific question comes from the following observation. In a system without spatial dynamics — thus without diffusion — a catastrophic transition occurs between sustained vegetation and bare soil when rainfall decreases below a critical point.

The transition point corresponds to a fold bifurcation, and can be qualified as a tipping point (Lenton et al., 2008). Rietkerk et al. (2021) suggested that spatial dynamics and the existence of the Busse ballon smoothens the transition between full-fledged vegetation and bare-soil. In their terms, the Busse Balloon 'evades' the tipping point. This result implies that spatial dynamics effectively lowers the precipitation threshold above which vegetation can be sustained.

Siteur et al. (2014) showed that a Busse balloon appears in the Klausmeier vegetation model (Klausmeier, 1999), where it occupies a region of the parameter space with lower rainfall than necessary to sustain a homogeneous vegetation. However, the nature of this transition through the Busse Balloon may be complex. For the non-spatial Rietkerk model, there is no such thing as a fold bifurcation. But we will see that adding a spatial component creates fold bifurcation.

The objective of the present study is to fill this gap, in the idealized context of a spatial domain of one dimension, by characterizing the intermediate states that may emerge during the transition from full vegetation to bare soil, and to examine the dynamics that underlie potential transitions between these states with a focus on a new type of equilibrium that we called "mixed state".

Specifically, we provide an in-depth analysis of the Busse balloon, demonstrate the co-existence of multiple equilibria for a given rainfall intensity, and foresee the circumstances which may trigger transitions between these different equilibria. In that sense, we propose an extension of the work by Zelnik et al. (2013) who computes partially the bifurcation diagram for Rietkerk's model. We also develop the method for finding equilibrium branches and characterize their stability. Finally, we highlight the existence of another type of equilibrium different than the regular patterns. We call them 'Mixed State' because of their shape and show how they can have an influence on the dynamics of the system.

## 2 Bifurcation diagram and stability

In this section, we present a method to construct the bifurcation diagram for Rietkerk's model and determine the form of the different equilibrium branches as a function of rainfall.

### Linear analysis and Turing zone

The classical approach is to consider as in Siero (2020), first, the static homogeneous equilibrium. To this end, we define the equilibrium $\bar{B} = B(x,t)$ with $\bar{B}$ a constant in time and space—likewise with the other variables—which satisfy the relationships:

$$0 = cg_{max}\frac{\bar{W}\bar{B}}{\bar{W} + k_1} - d\bar{B} \tag{2}$$

$$0 = \alpha O\frac{\bar{B} + k_2 w_0}{\bar{B} + k_2} - g_{max}\frac{\bar{W}\bar{B}}{\bar{W} + k_1} - r_w\bar{W} \tag{3}$$

$$0 = R - \alpha\bar{O}\frac{\bar{B} + k_2 w_0}{\bar{B} + k_2}. \tag{4}$$

Two solutions exist depending on the value of $R$. One solution with vegetation is:

$$\bar{B} = c\left(\frac{R}{d} - \frac{k_1 r_w}{cg_{max} - d}\right), \tag{5}$$

$$\bar{W} = \frac{dk_1}{cg_{max} - d}, \tag{6}$$

$$\bar{O} = \frac{R((cg_{max} - d)(cR + dk_2) - cdk_1 r_w)}{\alpha((cg_{max} - d)(cR + dk_2 w_0) - cdk_1 r_w)}. \tag{7}$$

$$\tag{8}$$

Physical equilibria for positive parameters must be positive, and hence satisfy the relations $cg_{max} > d$ and $R > k_1 r_w d/(cg_{max} - d)$. The homogeneous equilibria without vegetation are, by definition, $\bar{B} = 0$, which imply

$$\bar{W}_0 = \frac{R}{r_w} \tag{9}$$

$$\bar{O}_0 = \frac{R}{w_0\alpha}, \tag{10}$$

which are, again, valid for positive parameters. For the parameters chosen here, the homogeneous vegetated equilibrium exists for $R > 1$ and the non-vegetated equilibrium always exists [1]. Fig. 1 shows these equilibria as a function of rainfall $R$.

We now consider the stability of these equilibria. Again, following standard practice we consider spatially periodic perturbations as follows:

$$B(x,t) = \bar{B} + \epsilon\delta B(x,t), \tag{11}$$

$$W(x,t) = \bar{W} + \epsilon\delta W(x,t), \tag{12}$$

$$O(x,t) = \bar{O} + \epsilon\delta O(x,t), \tag{13}$$

with

$$\delta B(x,t) = \delta B e^{\Omega t + i\kappa x}, \tag{14}$$

$$\delta W(x,t) = \delta W e^{\Omega t + i\kappa x}, \tag{15}$$

$$\delta O(x,t) = \delta O e^{\Omega t + i\kappa x}. \tag{16}$$

---

[1]The universal existence of the non-vegetated equilibrium does not imply its stability for all values of $R$.

We now introduce the perturbation into the original equations, develop a Taylor expansion for small $\epsilon$ and get rid of the 0th order term by using the homogeneous equilibrium. The linearised equations obtained can be recast as an eigenvalue problem:

$$A = \begin{pmatrix} d - \frac{cg_{max}\bar{W}}{k_1+\bar{W}} + D_B\kappa^2 & -\frac{cg_{max}k_1\bar{B}}{(k_1+\bar{W})^2} & 0 \\ \frac{g_{max}\bar{W}}{k_1+\bar{W}} + \frac{k_2\alpha(w_0-1)\bar{O}}{(k_2+\bar{B})^2} & r_w + \frac{g_{max}k_1\bar{B}}{(k_1+\bar{W})^2} + D_W\kappa^2 & -\frac{\alpha(k_2w_0+\bar{B})}{k_2+\bar{B}} \\ -\frac{k_2\alpha(w_0-1)\bar{O}}{(k_2+\bar{B})^2} & 0 & \frac{\alpha(k_2w_0+\bar{B})}{k_2+\bar{B}} + D_O\kappa^2 \end{pmatrix} \begin{pmatrix} \delta B \\ \delta W \\ \delta O \end{pmatrix} = -\Omega \begin{pmatrix} \delta B \\ \delta W \\ \delta O \end{pmatrix} \tag{17}$$

Every pair of values $(R,\kappa)$ defines a new eigenvalue problem. It corresponds to the linear dynamics obtained by perturbing a homogeneous equilibrium that exists at a particular value of $R$ with wavenumber $\kappa$. We are interested in finding out whether there are values of $(R,\kappa)$ for which there exists an exponentially growing solution, thus positive $\Omega$. It is then said that the homogeneous equilibrium for a specific value of $R$ is linearly unstable to perturbations of wavenumber $\kappa$. For the eigenvalue problem to have a non-trivial solution, the determinant $det(A+\Omega)$ must be 0. This leads to a cubic equation for $\Omega$:

$$a_1(R,\kappa)\Omega^3 + a_2(R,\kappa)\Omega^2 + a_3(R,\kappa)\Omega + a_4(R,\kappa) = 0, \tag{18}$$

which we solve numerically on a physical domain defined by $\{R,\kappa\} \in \mathcal{R}$ with $R \geq 0$. This leads to three roots at each point in the domain, which we can refer to as $\Omega_1(R,\kappa)$, $\Omega_2(R,\kappa)$ and $\Omega_3(R,\kappa)$. Two of the roots, say $\Omega_{(2,3)}(R,\kappa)$, always have a real negative value, and hence correspond to exponentially decaying modes. One of the roots, $\Omega_1(R,\kappa)$, which is real in all the domain, has a positive real part in a region of the $(R,\kappa)$ plane. The countour of this area is shown in Fig. 2 by the black line. This defines the *Turing zone*. Hence, for those values of $(R,\kappa)$ with positive real part of $\Omega_1$ (inside of the contour) the corresponding homogeneous equilibrium is linearly unstable to inhomogeneous perturbations of wavenumber $\kappa$.

*Zero modes* correspond to marginally non-growing inhomogeneous solutions of the linearised equations and correspond to perturbations with $\Omega = 0$. Their existence usually signals the presence of an instability in the full non linear model, that can send the system towards an inhomogeneous time-invariant solution in the full nonlinear model. Specifically, Fig. 2 suggests the existence of two zero modes for any value of rainfall comprised between critical bounds $1.0 \, \mathrm{mm\,d^{-1}}$ and $1.25 \, \mathrm{mm\,d^{-1}}$. Such zero modes can satisfy the boundary conditions for a (low) wavenumber that depends on $R$, and therefore require a domain large enough to develop. If the domain is so small that its fundamental mode is in the stable region ($\kappa \gtrsim 0.6$), it is always stable. In this section, we found zero modes along the homogeneous branch compatible with a specific domain. Those zero modes indicate the start of inhomogeneous branches of equilibria. In the following section we will rely on those zero modes to compute equilibria to the full non linear model.

## 2.1 Bifurcation diagram and continuation method

The analysis in the previous section showed that homogeneous vegetation is linearly unstable against spatially periodic perturbations on the rain range $1 \, \mathrm{mm\,d^{-1}} < R < 1.25 \, \mathrm{mm\,d^{-1}}$. This linear analysis suggested the existence of periodic inhomogeneous equilibria in the nonlinear system within that rainfall range. Now, our focus shifts towards explicitly computing these nonlinear equilibria. To achieve this, numerical methods are employed to solve the equations involved.

For the time being we assume a periodic domain with a finite size of $L = 100$ m. The choice of the size of the domain influences the number of zero modes exhibited on the homogeneous equilibrium branch. This choice together with periodic

boundary conditions effectively discretize the set of zero modes (Fig. 3). Indeed, setting $\kappa = n\frac{2\pi}{L}$, with $n$ the wavenumber, we find two pairs $(\kappa, R)$ corresponding to zero modes, with the range $\kappa = [0 : 0.6]$, corresponding to perturbing the homogeneous equilibrium $\bar{S}(R)$ with a perturbation $\delta S(x) = \delta S \cos(\kappa x)$.

For each pair of $(\kappa, R)$ in the set of zero modes, we identify the corresponding homogeneous equilibrium $(\bar{S})$ and perturb it with the corresponding perturbation $\delta S \cos(\kappa x)$. The periodic boundary conditions are enforced by setting $\kappa = n\frac{2\pi}{L}$, with $n$ the wavenumber. The perturbed homogeneous solution

$$B_0(x, R) = \bar{B} + \delta B cos(\kappa x), \tag{19}$$

$$W_0(x, R) = \bar{W} + \delta W cos(\kappa x), \tag{20}$$

$$O_0(x, R) = \bar{O} + \delta O cos(\kappa x), \tag{21}$$

is then taken as the first guess input in a Newton-Raphson iteration used for finding the corresponding non linear inhomogeneous equilibrium.

At this point we need to discretize the spatial domain into, say, $N$ points. We used $N = 100$. Specifically, the discretized first guess reads $u_0(R) = [B_0(R), W_0(R), O_0(R)] \in \Re^{3N}$ and the algorithm

$$u_{i+1}(R) = u_i(R) - \epsilon J^{-1}(u_i(R)) f(u_i(R)), \tag{22}$$

converges towards the equilibrium $u_S(R)$ with $J(u_i(R))$, the Jacobian matrix of the system and $f(u_i(R))$, the right-hand side as in Eq. (1). The Laplacian is discretized with a periodic pseudospectral method. This equilibrium, $u_S(R)$, is then used as the first guess to solve for the equilibrium with rainfall $R + \delta R$,

$$u_0(R + \delta R) = u_S(R) + \delta u, \tag{23}$$

with $\delta u \ll \epsilon$. This iterative procedure leads to the construction of the full branch of equilibria.

A new code has been developed to implement this continuation method. This code is available online on the repository mentioned in the code availability section.

Each branch obtained by the above technique is denoted by $n$ between 1 and 9, with $n$ the wavenumber associated with the perturbation. The periodic boundary conditions, together with a finite domain size fixed the number of zero modes present within the Turing zone , and therefore, the number of inhomogeneous branches that exist. For the periodic L=100m domain, only nine zero modes fit within the Turing zone (see Fig. 3). Higher wavenumber modes correspond to linearly stable perturbations that exponentially decline.

All the branches obtained with this approach are shown in Fig. 4. We now see that the range of rainfall where vegetation is sustained is much wider than expected from the linear analysis. The linear analysis in the previous section predicted the growth of vegetation patterns in the Turing zone, between $1.0\,\mathrm{mm.d}^{-1}$ and $1.25\,\mathrm{mm.d}^{-1}$. The analysis of the full non linear system, presented in Fig. 4, actually reveals time-invariant pattern equilibria on branches attached to zero modes in the range $0.5 - 1.3$ $\mathrm{mm.d}^{-1}$. Although the non-linear equilibrium may differ in shape from the linear perturbation, it is found that wavenumber $n$ used to perturb the homogenous equilibrium describes reasonably well the shape of the pattern along the branch obtained from

that perturbation. One such equilibrium is displayed for illustration on the top panels of Fig. 5. For this particular equilibrium, with wavenumber $n = 2$, the biomass accumulates in two places, and soil water peaks there due to enhanced infiltration rate. Surface water accumulates around those areas.

## 2.2 Stability analysis

Characterizing the stability of equilibria will further help us to understand the dynamics of the system inside the Busse balloon. The Busse balloon is the region in the space parameter $(\kappa, R)$ where at least one stable pattern equilibrium exists. A linear stability analysis gives a partial but valuable information about a given solution and its possible evolutions. As we will see in the following, while stable equilibria tend to be the endstate of time evolutions, unstable equilibria can still be relevant for the dynamics. The stability of equilibria is classically estimated based on the Jacobian matrix evaluated at each equilibrium, positive real parts of the eigenvalues signaling instability.

As a first example, the equilibrium for a rainfall equal to $0.9 \, \mathrm{mm \, d^{-1}}$ of the branch $n = 2$, the associated eigenvalues and the first eigenvector are displayed on Fig.5. The majority of eigenvalues have a negative real part. One eigenvalue has a very small, positive real part ($3 \cdot 10^{-10}$), which suggests quasi-neutral stability. Further inspection of the associated eigenvector shows that this quasi-neutrally stable mode is the spatial derivative of the equilibrium pattern. Therefore, it corresponds to a translation mode, which is indeed expected with periodic boundary conditions. Hence, we have a stable pattern which may however translate consistently with the periodic character of the boundary conditions. We therefore attribute the small real part to a numerical artifact (it should be zero) and consider equilibria with positive real part of the highest eigenvalue of the order of $10^{-10}$ as stable.

This stability analysis is repeated for all the equilibria shown in Fig. 4. Stable sections of the branches are drawn in solid lines, and unstable ones with dashed lines. This gives us a more precise idea about what is happening inside the Busse balloon. We now know what is the shape of those stable states, and how and at what rainfall value they lose stability. The existence of multiple stable states for a given value of rainfall is also consistent with previous work based on models and real systems (Bel et al. (2012), Bastiaansen et al. (2018))

Some branches of equilibria, such as $n = 5, 6, 7, 8, 9$ are unstable across all their existence domain. Other branches, as $n = 1, 2, 3, 4$, change stability along the branch. These branches of equilibria, start as unstable from the low vegetation zero modes on the homogeneous equilibrium. Then, they become stable, and eventually become unstable again shortly before joining the highly vegetated homogeneous equilibrium at the corresponding zero mode. The change in stability may take place at the extreme rain values for which the equilibrium exists, as in the $n = 1$ branch, or at a rainfall value in the middle of a branch.

As we know, when a equilibrium loses stability, the real part of one of its eigenvalues changes from negative to positive, which indicates that the equilibrium becomes linearly unstable in the direction of the associated eigenvector. In all of the cases here the change of sign of the eigenvalue happens through zero (as opposed to, through infinity), meaning that, there are zero modes at the intersection of the stable and unstable branch sections. As we show next, this indicates the branching of another branch of equilibrium of the full-non linear system. A simple way to identify those zero modes is proposed in Appendix B.

## 2.3  "Mixed states"

In the previous section we studied the stability of the "main" branches of equilibria shown in Fig. 4. These branches of equilibria originated from the zero modes present on the homogeneous vegetated branch. We also saw that some of these branches of equilibria had zero modes at the intersection of their stable and unstable parts. These zero modes act as bifurcation points from which new branches of equilibria emerge. The latter can be found by starting from the equilibrium at the bifurcation point, perturbed in the direction of the eigenvector associated with the newly positive eigenvalue (zero mode) corresponding to a new unstable direction. Again, a Newton-Raphson iteration allows us to find the new equilibrium. From there, the continuation method explained in the previous section allows us to trace the full new branch.

The result of this routine applied to the bifurcation point of the branch $n = 4$ is shown on the upper panels of Fig. 6. At the transition between stability and unstability, there are three zero modes. Hence three branches of equilibria start at this transition and they are shown in Fig. 6. They begin all at the transition between the stable and unstable part of the $n = 4$ branch and they reconnect with it for higher rainfall. We call those branches $n = 4\,bis_1$, $n = 4\,bis_2$ and $n = 4\,bis_3$. Those three branches are close (regarding to the mean biomass) to that of the $n = 4$ branch but if we look closely at their profile (bottom panels of Fig. 6) we see how they differ. For a given value of rainfall, the mixed state equilibria look like a modulation of the $n = 4$ equilibrium by an other wavenumber.

Equilibria branching out of zero modes in the homogeneous equilibrium tend to exhibit a single perturbation mode, see Fig. 5. By contrast, equilibria branching out of zero modes in those inhomogeneous equilibria exhibit a mixture of modes (lower panels of Fig. 6), hence the name "mixed states". The equilibria along the mixed state branch $n = 4\,bis_3$ are unstable, with positive eigenvalues. The branches $n = 4\,bis_2$ and $n = 4\,bis_3$ reconnect around $1.14\,\mathrm{mmd}^{-1}$. There are numerous mixed state branches, obtained from the different zero modes at bifurcation points found along the main branches. These can also be found to emerge from zero modes present on the unstable sections of the main branches. For example, in the low vegetated unstable part of the $n = 2$ branch (see Fig. A2), we find two zero modes at the same value of rainfall, from which two mixed state branches emerge: one labelled $n = 2\,bis$ because it connects to the $n = 2$ branch and one labelled $n = 3\,loc$ which connects to the $n = 4$ branch. The $n = 3\,loc$ refers to the fact that the equilibrium of that branch is localized in space. The $n = 1$ main branch is special in two ways. First, as Zelnik et al. (2013) stated, it is the only stable localized state. Second, it starts from a zero mode on the homogeneous equilibrium, but it ends up connecting to the $n = 2$ branch as if it was a "mixed state", see Fig. A1. The evolution of the shape along this branch goes from 1 localised vegetation peak to 2 localised vegetation peaks in the unstable part close to the connection with the $n = 2$ branch.

All mixed state branches share similar characteristics, with states appearing as a modulation of a main branch. Even though they are unstable, we expect mixed state equilibira to influence the dynamics of nearby trajectories, depending on the value of the positive eigenvalue.

# 3 Numerical simulations of trajectories

To assess relevance of the bifurcation diagram for understanding transient dynamics, we performed two series of numerical simulations.

## 3.1 Trajectories in a changing environment

In order to assess the relevance of the various branches for the dynamics, we propose the following numerical experiment. The Rietkerk model is run with a rainfall changing over time with a rate of $\frac{dR}{dt} = -5.10^{-6}\,\mathrm{mm\,d}^{-2}$. The starting point is a the vegetated homogeneous equilibrium with rainfall $R = 1.4\,\mathrm{mm\,d}^{-1}$. The final rainfall is $0.5\,\mathrm{mm\,d}^{-1}$. The result is shown in Fig. 7. Until $R \sim 1.2\,\mathrm{mm\,d}^{-1}$, the system stays in the homogeneous equilibrium. At this point, the system undergoes a Turing bifurcation and jumps to an heterogeneous equilibrium. The patterned branch on which the system lands is the $n = 2$ branch. We can understand this transition by looking at Fig. 3. Indeed we see there that the first mode to destabilize in a decreasing rainfall scenario is the $n = 2$. So this simple linear analysis can give us a first idea about the way the system will be destabilized in a slow change scenario. After that, the system tracks the $n = 2$ branch until this equilibrium loses its stability. Finally, the system switches to the only other stable equilibrium, the $n = 1$. For this experiment, the mixed states don't have an influence on the trajectory. But in the following we will see that even if mixed state are unstable, they still can play a role in the dynamics.

## 3.2 Effect of a "Mixed State" on trajectories

Here we address the question of whether unstable mixed state equilibria are also able to influence system transient trajectories. In this case the rainfall is set to a rainfall $R = 1.05\,\mathrm{mm\,d}^{-1}$, for which a mixed state equilibrium exists.

The initial condition is an unstable equilibrium, $n = 5$, with a small perturbation along the direction of the first eigenvector. The top panels of Fig. 8 summarize the evolution of the biomass pattern over time and the corresponding trajectory in a summary phase space. This summary phase space consists of a two dimensional phase space where the two dimensions are the mean biomass and the maximum biomass. With that in mind, we observe that the system leaves the $n = 5$ unstable equilibrium by first reorganising itself: mean biomass remains constant while maximum biomass increases. This means that one or more vegetation bumps are growing at the expense of others. After that initial reorganisation, an excursion on higher mean and maximum biomass takes place. It is during this out of equilibrium excursion, that one of the vegetation bumps is lost. At this point in time the vegetation profile, containing four vegetation bumps, passes close to the three mixed states originating from zero modes on the $n = 4$ branch explained on the previous section. Then, the system undergoes an other transition to the $n = 3$ equilibrium. To have a better representation of this trajectory, we also show on the upper right panel of Fig. 8 the time evolution. We see that the system quite slowly leaves the $n = 5$ unstable equilibria. The state is indeed unstable, but the associated positive eigenvalues are small, such that the dynamics around the equilibrium are slow. After abruptly departing from the $n = 5$ equilibrium, one biomass bump disappears, leading to a rearranged state with four bumps.

We see that even though the summary phase space may let us think that the system passes close to the mixed state $n = 4bis_2$, we have to keep in mind that this two dimensional space is obtained by the projection of the infinite-dimensional phase space

on a two dimension summary space. On the three lower panels are represented the biomass profile at three points in time, and their corresponding positions are also shown on the left upper panel. On each of the lower panels the shape of the $n = 4bis_3$ mixed state is also shown. The shape of the dynamical solution is actually close to the $n = 4bis_3$. That equilibrium is unstable, but the system lingers in its vicinity for a considerable period of time before another bump of biomass vanishes, propelling the system towards the stable equilibrium of $n = 3$.

Switching from one state to another by losing one or more vegetation bumps is a known feature of vegetation pattern model (Bastiaansen and Doelman (2019), Bastiaansen et al. (2020)). This mechanism is called sideband instability (Doelman et al. (2012), Siteur et al. (2014)). We used the same setup with different rainfall values along the $n = 5$ branch. The fact that the system passes close to a mixed state is consistent along the $n = 5$ branch only for values of rainfall for which mixed state equilibria exist. For rainfall lower than $\sim 0.91 \mathrm{mm\,d}^{-1}$, the system jumps directly to the $n = 3$ equilibrium. Now if we consider the same rainfall but we start from an other unstable branch like the $n = 6$ or $n = 7$, the dynamics is different. For $n = 6$, we observe a different type of destabilization called the period-doubling (Doelman et al. (2012), Siteur et al. (2014)). With that mechanism the system transitions to the $n = 3$ equilibrium without passing through a mixed state. And the system spends less time around the unstable equilibrium of $n = 6$ compared to the $n = 5$ case. This might be explained by the fact that the positive eigenvalue on the $n = 6$ solution is larger than the one on the $n = 5$ solution. For $n = 7$, the transition is even more rapid, as expected for an even larger positive eigenvalue, but the landing point is still $n = 3$. In that case, we observe a destruction of four bumps similar to a sideband instability.

## 4 Discussion

We uncovered the existence of mixed states in Ritkerk's model and showed their importance for transient dynamics. As far as we are aware, such mixed states have never been identified nor described in models for vegetation patterns.

Mixed states emerge at the transition between unstable and stable states along a branch of equilibria, and have a functional form that appears as the combination of two equilibria from the main branches. We found that while these equilibria are unstable, they may still influence the system's dynamics by slowing down its evolution when it passes near them.

The influence of unstable modes on dynamics has been studied in ecological models. For example Sherratt et al. (2009) showed how spatio-temporal chaos appears in the wake of an unstable wavetrain for the complex Ginzburg-Landeau equation. In the more general context of ecology, Hastings et al. (2018) and more recently Morozov et al. (2020) proposed a classification of transient phenomena in ecological models based on a dynamical system approach. In their definition, a dynamical regime is transient if it persists for a sufficiently long time (quasi-stable) and if the transition between two regimes occurs on a much shorter timescale than the time of existence of the quasi-stable regime. According to that definition, the behavior observed in section 3.2 can be seen as a transient phenomenon. Indeed, the systems spends hundred to thousands of days around unstable states($n = 5$, $n = 4bis_1$ and $n = 4bis_3$) and then takes only a few days to jump to the final stable equilibrium. By following the typology presented in those two papers, the transient observed here can be qualified as a 'crawl by' transient. Indeed, the transient observed is linked to the existence of a saddle-type invariant set: the mixed state. Transient dynamics have also been

studied in the context of coexistence in vegetation patterns by Eigentler and Sherratt (2019) and are characterized by the small size of a positive eigenvalue. Long transients can also be observed in very slow front invasion dynamics (Van De Leemput et al., 2015) The mechanism behind the transient is different but the effect is the same: the system spends a long time in a region of the phase space without attractor.

As we see in section 3.2, the history and the initial conditions of the system are important to see whether the mixed state will appear or not in the dynamics Other vegetation patterns models exhibit this type of sensitivity to history and initial conditions (Sherratt (2013), Adams et al. (2003) and Alberti et al. (2015)). Further research is needed to determine whether mixed states are a common feature of reaction-diffusion systems.

We adopted periodic boundary conditions, as in previous studies (Rietkerk et al. (2002), Dekker et al. (2007)). As stated by Dijkstra (2011), the periodic boundary conditions allow for the existence of unstable equilibria with high wavenumber. However, this boundary condition, widely used in reaction-diffusion models due to its simplicity and convenience, may not reflect real-world scenarios accurately. More generally, working with a periodic domain discretizes the set of admissible equilibria. Hence, the significance of mixed states for describing the dynamics of non periodic infinite-size systems needs be further assessed.

As a step towards this objective, we computed the bifurcation diagram for a larger domain size of $L = 200 \, \text{m}$, instead of $L = 100 \, \text{m}$. As expected, we observed more branches in this case but, remarkably, the branches and the stability of the branches shared between the two domains are the same as for $L = 100 \, \text{m}$. Of course the even $n$ number branches for the $L = 200 \, \text{m}$ domain are the same that the $L = 100 \, \text{m}$ domain. The new branches are all designated with odd $n$ number as a consequence of the extended domain. The range of rainfall that can support a stable vegetation pattern also increased slightly with the larger domain size, with the $n = 1$ branch extending this range.

These findings suggest that the existence of mixed states and their stability properties is robust across domain sizes, even though how they would manifest themselves in continuous large domains is still to be established.

## 5   Conclusions

We present an in-depth analysis of the Rietkerk model's behavior that we believe sheds a new light on the dynamics of vegetation patterns. Our analysis is based on the bifurcation diagram of the vegetation pattern model, which we constructed using a variety of techniques. First, we performed a linear stability analysis of the homogeneous equilibrium of the system, which allowed us to delineate the so-called Turing zone. This zone is characterized by the instability of the homogeneous equilibrium to small heterogeneous perturbations, and it serves as a starting point for our analysis. Next, we used perturbed states at the edge of the Turing zone, the so-called zero modes, to construct whole branches of equilibria associated with the system. To obtain these branches, we used a continuation method based on the Newton-Raphson algorithm. By doing so, we were able to obtain the heterogeneous equilibria for a given size of the domain. As expected, due to the system's non linearity, we discovered that the system allows for multiple equilibria for a given value of parameter, specifically rainfall. We went a step further by assessing the stability of those equilibria by computing the eigenvalues and associated eigenvectors of the Jacobian

matrix. This stability analysis showed that inhomogeneous vegetated states can exist and be stable, even for values of rainfall for which no vegetation exists in the non-spatialized system. Yet, we also found that the main branches of equilibrium originating from a zero mode are not the only ones present in the system. At the transition between unstable and stable states along one branch, a new type of equilibrium appears. The latter, which we called "mixed state", are always unstable equilibria and they look like a mix of two equilibria from the main branches. Although these equilibria are unstable, they affect the dynamics by slowing down the evolution of the system when it passes close to it. This slowing effect has been described in ecology, but to the best of our knowledge, the mixed states influence on the dynamics had not been previously shown for vegetation patterns. Overall, our approach allowed us to construct a bifurcation diagram that gives us valuable insights about the behavior of the system. This approach is helpful to disentangle the fate of the system in the Busse balloon and could be used to assess the existence or not of mixed states in spatially extended systems.

*Code availability.* All the code used to produce the figures for this manuscript are available here: https://github.com/lvanderveken/Rietkerk_bif_diag

## Appendix A: Random initial conditions with a fixed rainfall

Here, our objective is to verify that the stable pattern branches act as attractors in the dynamics of the system. To this end we specify a value of rainfall for which multiple stable equilibria exist. Starting with different initial conditions and evolving the system, we expect that different runs end in some of the different available stable states for the chosen rain value. To avoid favoring the attraction to a particular branch we opt for bare soil initial conditions with random noise. Rainfall is set to $R = 1.1\,\mathrm{mm\,d^{-1}}$, for which the system has several stable equilibria (n = 2,3 and 4). The model is run for 5000 days with 500 distinct random initial conditions. Those are random from a uniform distribution (positive) noise on top of a homogeneous bare soil equilibrium. Trajectories, projected onto a summary space (mean and maximum biomass) are shown in Fig. A4. As this is a summary space—a projection of the full space—trajectories may appear to cross, but they do not in the full space. Out of the 500 runs, 465 land on the $n = 2$ equilibrium, and the others land on the $n = 3$ equilibrium. This indicates that the equilibria that we identify as stable are indeed those which attract trajectories. Having no trajectories on the $n = 4$ equilibrium suggests that the basin of attraction of the latter is narrow close to the homogeneous bare soil equilibrium, with few chances for a trajectory starting from random initial conditions to be in the basin of attraction of that equilibrium or, in more informal terms, to evolve such as to pass close enough to the $n = 4$ equilibrium and land on it. The $n = 4$ equilibrium can however be reached with carefully designed initial conditions, such as a cosine with a wavenumber equal to $n\frac{2\pi}{L}$.

## Appendix B: Identification of zero modes along branches

In section 2.3, we showed that new type of equilibrium (mixed state) can appear at the transition between unstable and stable part of a branch. This transition is linked to the transition from positive to negative of eigenvalue. For example at the transition

on the $n = 4$ branch, we have three zero modes leading to the appearance of three mixed state branches. In figure A3, we
propose a way to identify and represent those zero mode. Each of the panel are linked to one of the four highest eigenvalues.
If this eigenvalue is positive the line is dashed and if the eigenvalue is negative the line solid. With that in mind, let's focus
on the $n = 4$ branch, we see that around $R = 0.91 \, \mathrm{mm \, d^{-1}}$ the three first eigenvalues become negative. This means that there
are three zero modes. We also observe that for the $n = 3$ branch, two eigenvalues switch from positive to negative around
$R = 0.73 \, \mathrm{mm \, d^{-1}}$. Again, this means that two mixed state branches emanate from the zero mode at the transition.

*Author contributions.* LV, MC and MM designed the study. LV and MM performed the numerical analysis. LV wrote the manuscript and
MC and MM reviewed and edited the paper.

*Competing interests.* The contact author has declared that none of the authors has any competing interests

*Acknowledgements.* This project has received funding from the European Union's Horizon 2020 research and innovation programme under
grant agreement no. 82097. Michel Crucifix is funded as Research Director by the Belgian National Fund of Scientific Research. The authors
use chat-GPT3.5 to help rephrasing some parts of the manuscript.

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

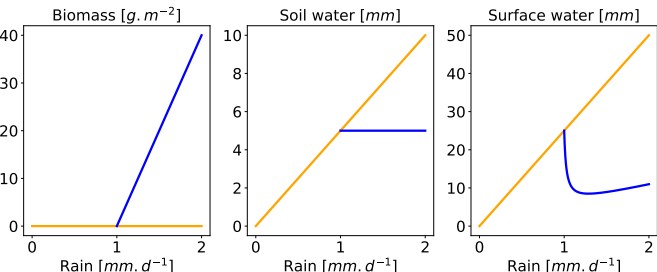

**Figure 1.** Homogeneous equilibria of Rietkerk's model. From left to rigth are represented the biomass, the soil water and the surface water. Blue line is the equilibrum with vegetation, yellow is without.

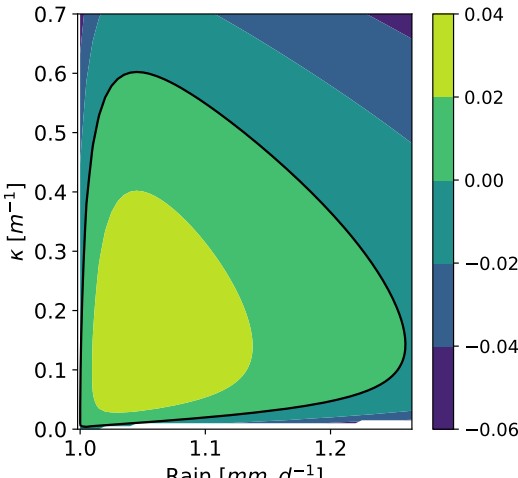

**Figure 2.** Contour plot for $\Omega_1(R, \kappa)$. The thick black line is the contour $\Omega_1 = 0$ and the region inside that contour corresponds to the parameter region $(R, \kappa)$ which in which the homogeneous equilibria is linearly unstable against inhomogeneous perturbations, i.e., $\Omega > 0$.

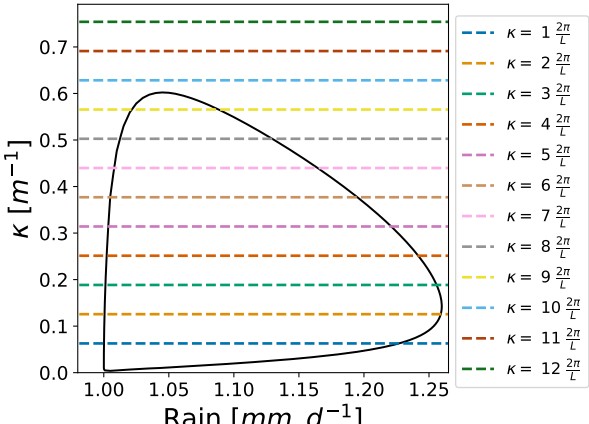

**Figure 3.** Zero modes present in a domain of $L = 100\,m$. Horizontal lines correspond to harmonics that fit inside the periodic $L = 100$ domain. The intersection of the horizontal lines with the $\Omega_1 = 0$ contour, provides the values of rain $R$ at which new branches of equilibria might appear. Notice that each of the 9 horizontal lines that intersect the contour, intersect it twice, hence 18 zero modes.

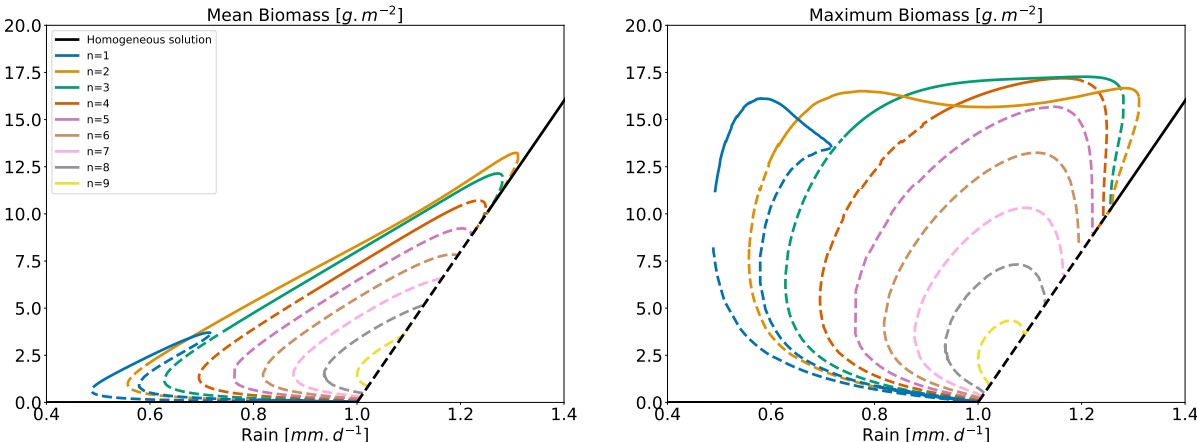

**Figure 4.** Bifurcation diagram for the Rietkerk's model with $L = 100\,m$. Each branch is labeled by an integer corresponding to the order of the wavenumber associated with the zero mode. Solid lines correspond to stable states and dashed lines to unstable states.

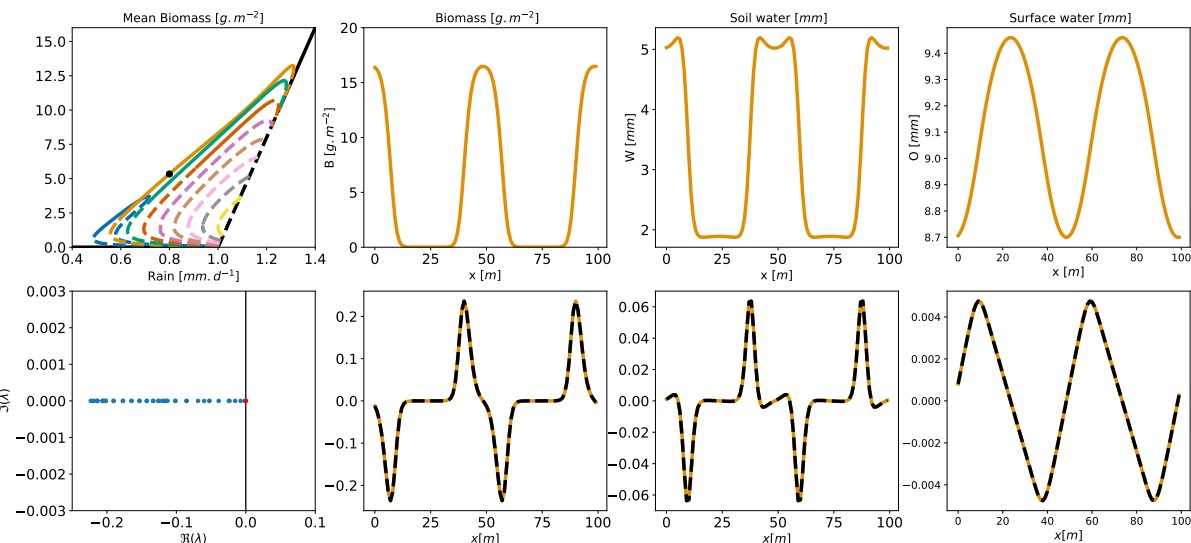

**Figure 5.** On the upper panels, equilibrium for a given $R$ (rainfall) $(0.9\,\mathrm{mm\,d^{-1}})$ on the $n = 2$ branch is shown. The black dot on the left-hand-side panel shows the position of the equilibrium on the bifurcation diagram. The three other panels show, from left to right, $B$ (biomass) $[\mathrm{g\,m^{-2}}]$, $W$ (soil water) [mm] and $O$ (surface water) [mm]. On the left-hand-side lower panel, eigenvalues of the Jacobian matrix associated with the equilibrium. The real part is on x axis and the imaginary part on the y axis. The larger eigenvalue is represented with a red dot. The black vertical line marks the $0$ position. The three other lower panels show, from left to right, the eigenvector associated with the larger eigenvalue projected on $B$ (biomass), $W$ (soil water) and $O$ (surface water) and with a dashed black line, the first spatial derivative of the equilibrium rescaled.

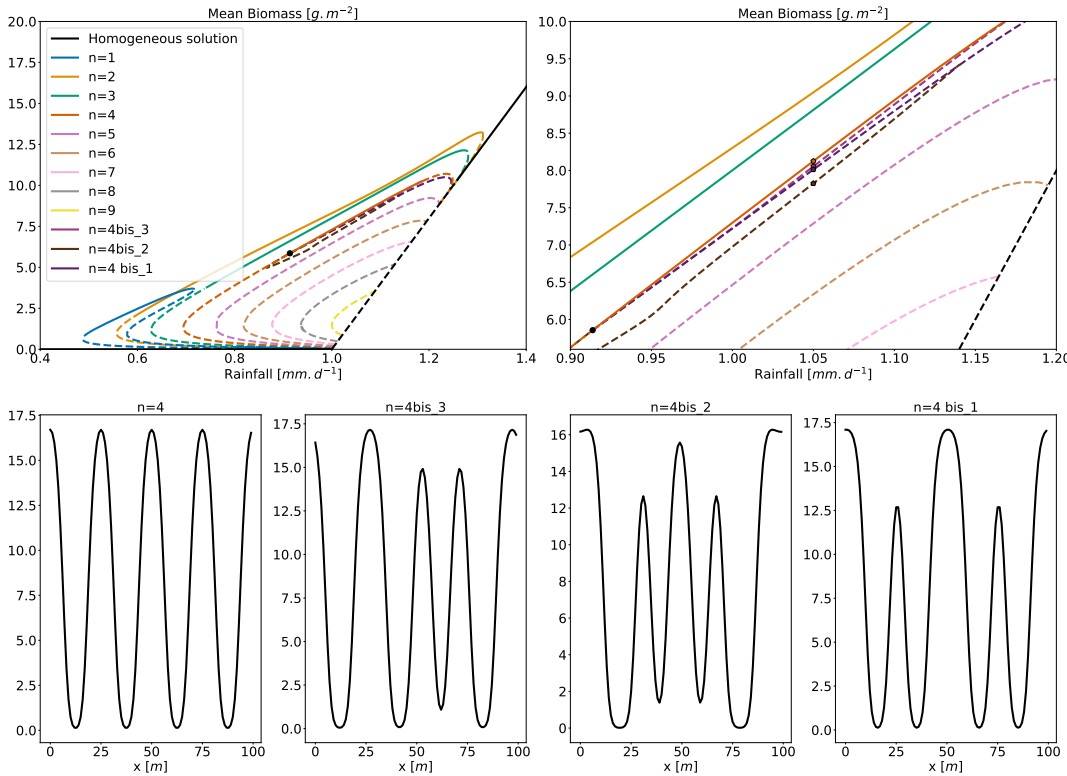

**Figure 6.** On the left-hand-side upper panel, bifurcation diagram with the addition of mixed state branches. The mixed state branches originating the zero modes along the $n = 4$ branch are called $n = 4bis_1$, $n = 4bis_2$ and $n = 4bis_3$. The bifurcation point is marked with a black dot. On the right-hand-side upper panel, an enhancement of the area around the bifurcation point. On the lower panels, the biomass profile of the $n = 4$ and the mixed states for a particular value of rainfall. The position on those equilibria on the bifurcation diagram are marked by circle on the right-hand-side upper panel

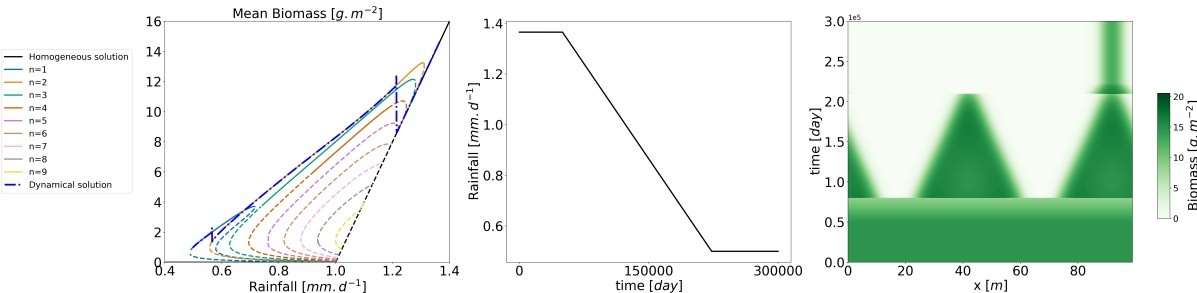

**Figure 7.** Trajectory of the Rietkerk model under a decreasing rainfall scenario. On the left panel, the trajectory in the bifurcation diagram plotted in blue dashed-dot line. On the middle panel, the rainfall with respect to time and on the right panel, the time evolution of the biomass.

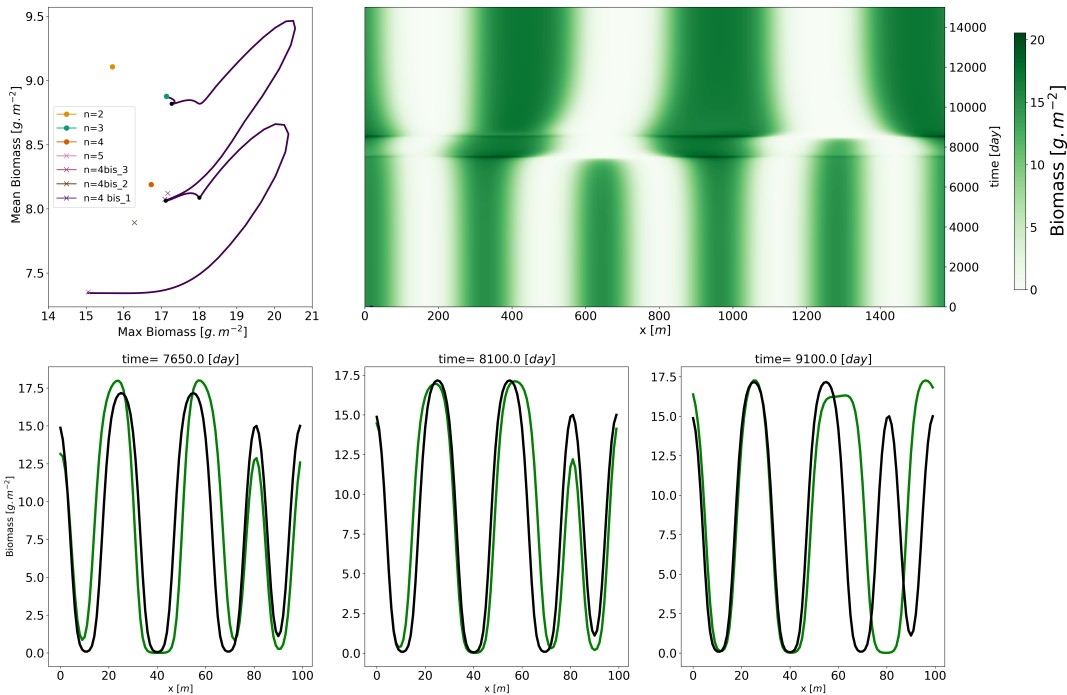

**Figure 8.** On the left upper panel, trajectories on a summary phase space from an initial condition close to the $n = 5$ equilibrium with a fixed rainfall $R = 1.05\,\mathrm{mm\,d^{-1}}$. The relevant equilibria are shown, the stable equilibria are represented with a circle and the unstable equilibria with a cross. On the right panel, we show the time evolution of the biomass. On the three lower panels, the dynamical solution at three different times is represented (in green) and also the shape (in black) of the mixed state denoted by $n = 4bis_3$.

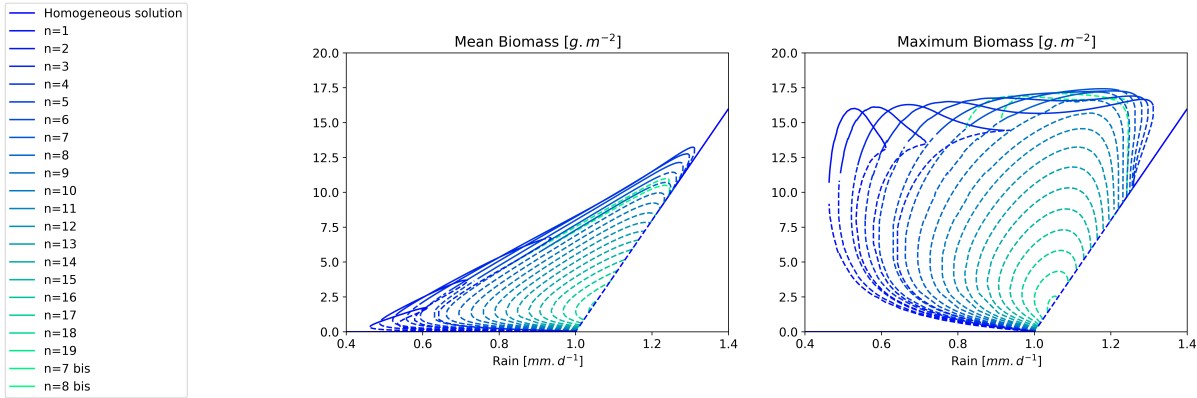

**Figure 9.** Bifurcation diagram for the Rietkerk's model with $L = 200\,\mathrm{m}$. Each branch is labeled by an integer corresponding to the order of the wavenumber associated with the zero mode. Plain lines correspond to stable states and dashed lines to unstable states.

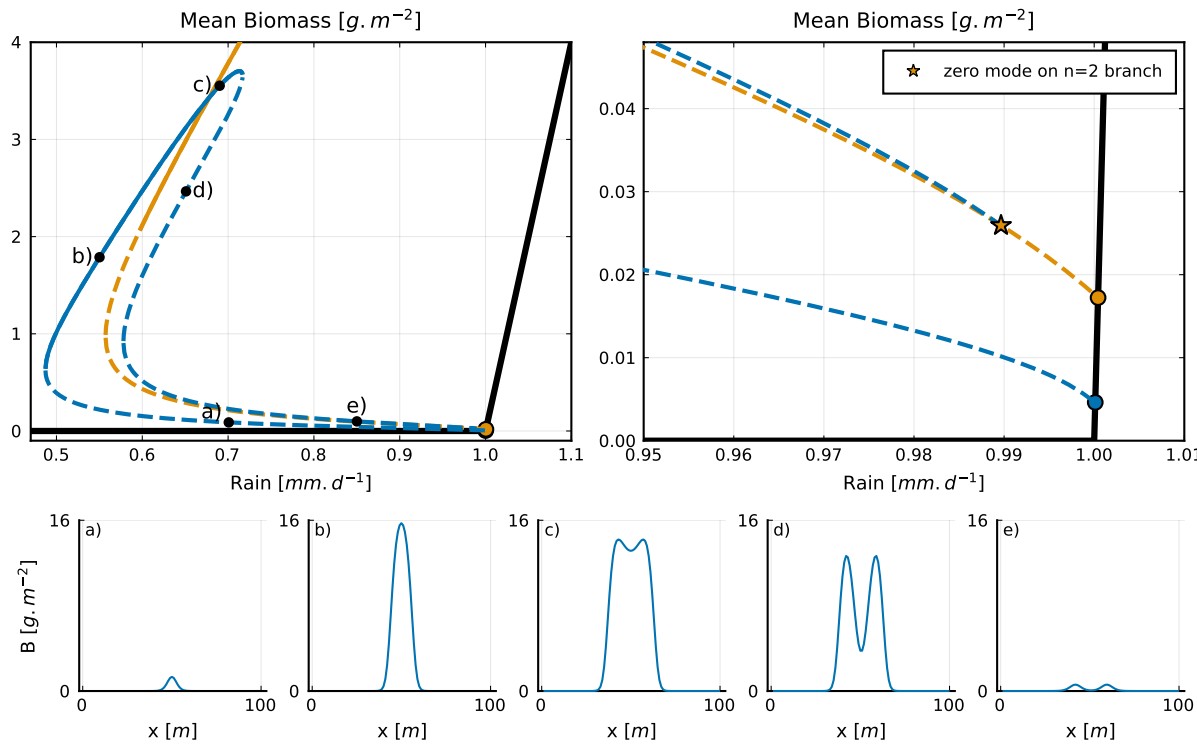

**Figure A1.** A focus in the bifurcation diagram on the $n = 1$ and $n = 2$ branches; the stable (unstable) states are noted with a (dashed) line. On the left-hand side upper panel, a zoom in the area of the bifurcation diagram where the $n = 1$ branch connects with the $n = 2$ branch. This connection is represented by a star. On the lower panels, equilibria along the $n = 1$ branch are shown. Each corresponds to a black dot in the right-hand side upper panel.

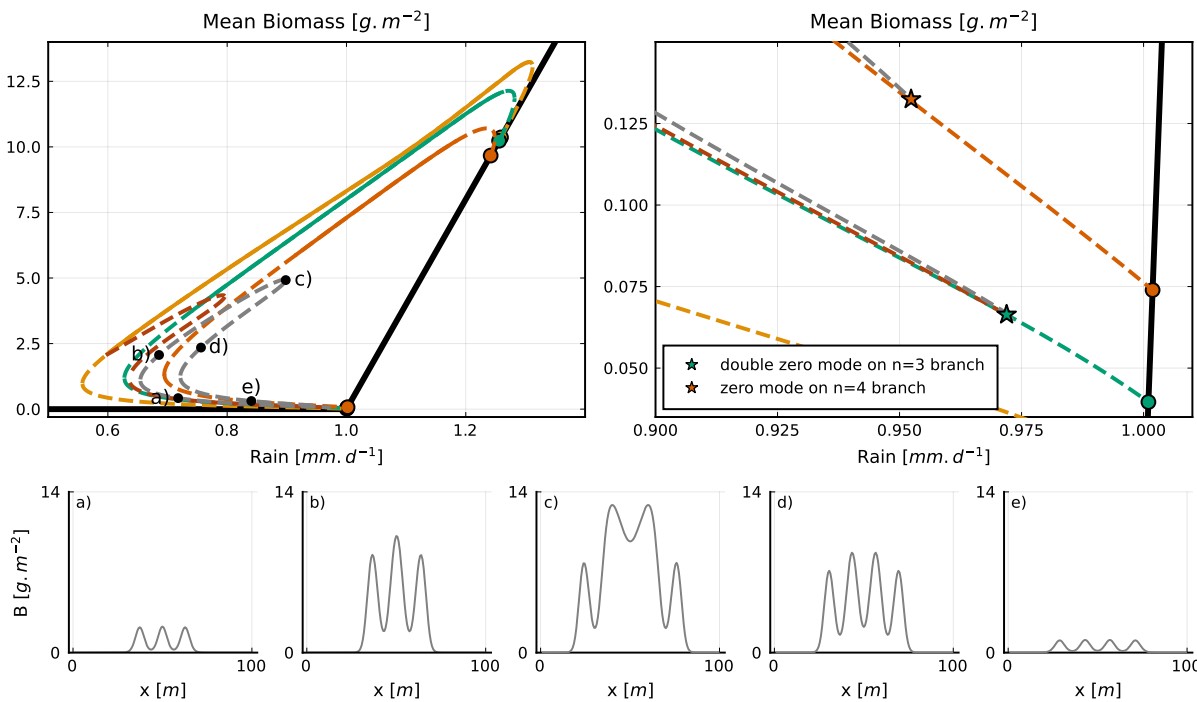

**Figure A2.** A focus on the bifurcation diagram of the $n = 2$, $n = 2bis$, $n = 3$, $n = 3loc$ and $n = 4$ branches; the stable (unstable) states are noted with a (dashed) line. On the left-hand side upper panel we show a zoom of the bottom part of the bifurcation diagram. The $n = 3$ branch exhibits a double zero-mode, indicated by the green star. Two equilibria emerge from that double-zero mode, the $n = 2bis$ and the $n = 3loc$. The $n = 3loc$ branch is characterized by a three bumps solutions which are localized. The first one, connects with the $n = 2$ branch while the second one connects with the $n = 4$ branch at the zero mode indicated by the orange star. On the lower panels, equilibria along the $n = 3loc$ branch are shown. Each corresponds to a black dot in the right-hand side upper panel.

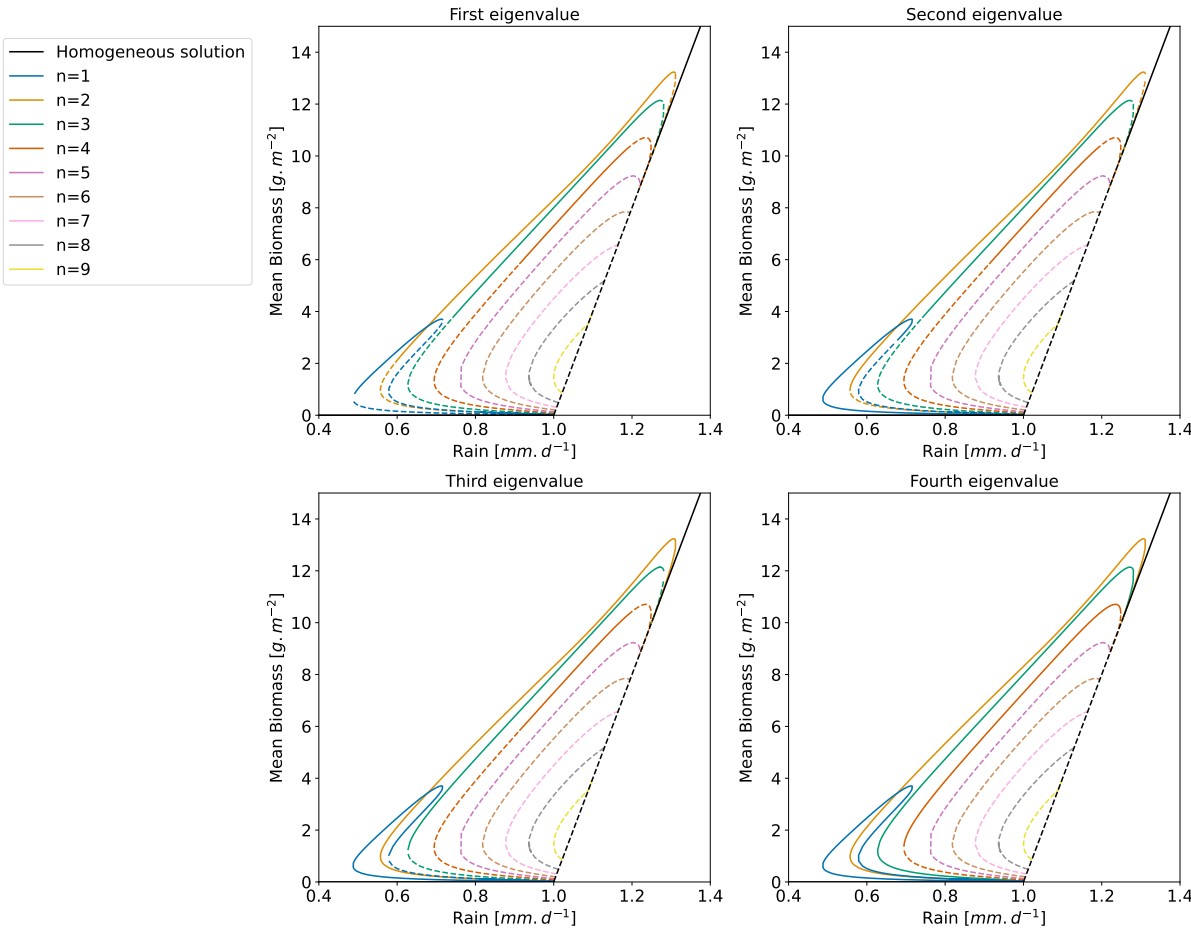

**Figure A3.** Bifurcation diagram for which the stability is computed based on the four highest eigenvalues. The solid line means that the eigenvalue considered is negative. The dashed line means that the eigenvalue is positive

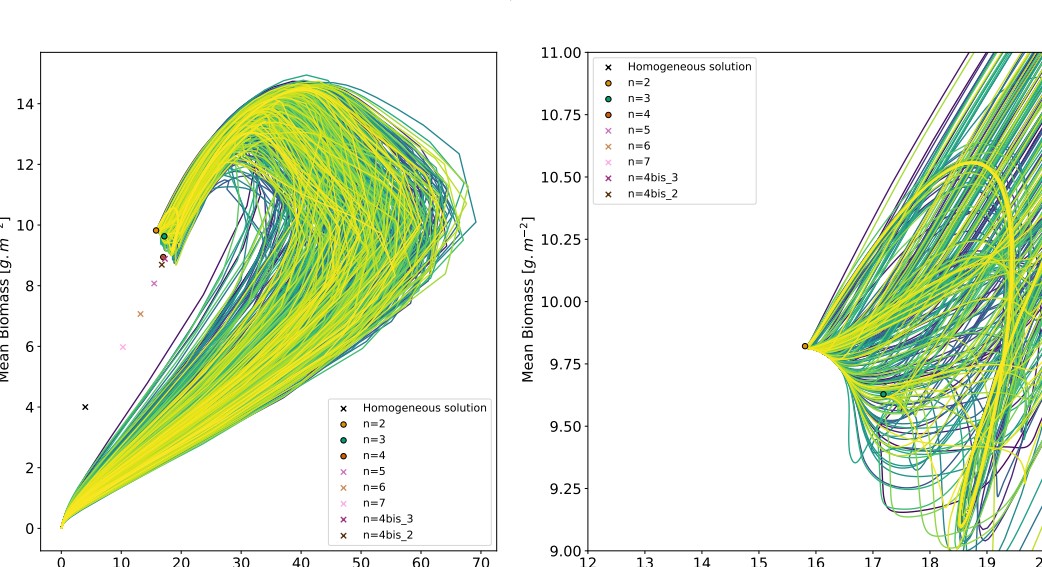

**Figure A4.** Trajectories on a summary phase space from random initial condition with a fixed rainfall $R = 1.1\,\mathrm{mm\,d^{-1}}$. The relevant equilibria are shownthe stable equilibria are represented with a circle and the unstable equilibria with a cross.