# Peer review of "Existence and influence of mixed states in a model of vegetation patterns"

_EGUsphere, 2023_

## Referee Comment (RC3)

This manuscript discusses the Rietkerk model for vegetation patters, based on a system of nonlinear equations for biomass, soil water and surface water. Various stable and unstable solutions are found, with a special focus on the existence of mixed states, which appear at the transition between stable and unstable solutions.

In my opinion, the manuscript is interesting, providing numerical solutions which allow to understand the essential features of the system. However, part of the discussion is not clear, some of the figures are not consistent with the text, and sometimes notation is confusing, so the manuscript should be improved before being considered for publication.

My detailed comments on the text are below:

1. Page 2, before Eq. (1). In the system of equations, all quantities should be defined ($c$, $g_{max}$, $k_1$, etc.).

2. Page 3, paragraph 1: "how they can have an influence on the dynamics of the system". Is it correct to say that a solution can have an "influence" on the dynamics of the system? Because a solution is an expression of the dynamics itself, it does not *influence* the dynamics.

3. Page 3, after Eq. (10): "Fig. 01 shows these solutions as a function of rainfall $R$". The description in the text seems to correspond to the leftmost panel, with a vegetated solution for $R > 1$. But what do the other panels represent?

   Also, the second and third panels show yellow lines (solutions without vegetations) with biomass larger than zero. How this should be interpreted?

4. Page 4, Eq. (18). It is confusing to use letters that have been used before ($c$, $d$). Please use other names for these variables.

5. Page 5, paragraph 2: "$\delta = \cos(\kappa x)$". This may be confusing, as $\delta$ has other meanings elsewhere, please change.

6. Page 5, after Eq. (23): "This iterative procedure leads to the construction of the full branch of solutions." Could there be other nonlinear solutions, which cannot be obtained via this method? (Because they are not in the same attraction basin of the Newton algorithm, for instance.)

7. Page 6, paragraph 4: "displayed on Fig.05." I do not think I understand this figure. The matrix has dimension 3, so for a given value of $R$ and $\kappa$, there should be 3 eigenvalues. Why are there so many points in the lower left panel of Fig. 5?

   What does the black line represent? Is it there only to mark the 0 position on the horizontal axis? This should be explained. Also, it should be explained what are the vertical and horizontal axis (imaginary and real part?).

8. Page 6, paragraph 4: "$3.10^{-10}$". Since eigenvalues represent frequency, do they have units? Other quantities in the system of equations do have units ($R$, $B$, etc.).

If not, how is frequency normalized?

Also, please change the notation to use a centered dot (`\cdot`).

9. Page 7, line 203: "we expect mixed-state modes to influence the dynamics of nearby trajectories, depending on the value of the positive eigenvalue.". This should be clarified. How can one mode "influence" the dynamics of other orbits, and what is the role of the positive eigenvalue in this.

10. Page 8, line 219: "the basin of attraction of the latter is narrow". Or it could be that the basin of attraction does not include regions close to the homogeneous zero state. Maybe starting from other points in phase space leads to $n = 4$, and the basin of attraction may turn out to be not small.

11. Page 8, line 220: $s = 4$ should be $n = 4$?

12. Page 8, line 224: "unstable mixed-state solutions are also able to influence system transient trajectories." It is not clear that Fig. 8, which is used to explain this, really settles this. It shows that starting from an unstable mixed-state leads to an interesting dynamics. On the other hand, one should always expect that starting from an unstable solution leads to a transient trajectory, until the system is attracted to a stable solution.

So it is not clear that the numerical example actually shows that mixed-state solutions "influence" trajectories. This should be better explained, and maybe rewording the statement is enough. (I am not sure that "influence" is the right word here.)

13. Page 9, table 1. Should it be "$9 \cdot 10^{-9}$" in the third column?

14. Page 9, caption of table 1: "for two unstable states". It should have some reference to Fig. 08, as curves for a given value of $n$ are not the same for different conditions.

15. Page 9, line 244: "unstable modes on neighbouring dynamics is well-known in complex systems (Lucarini and Bódai, 2017)". If I understand correctly, the reference provided shows that what they call an edge state can, for instance, connect to coexistent attractors via noise induced transitions. But the authors here do not discuss the role of noise to connect two stable modes (not in Fig. 8 at least), so, again, I am not sure the authors have been clear on their claims regarding this statement.

16. Page 9, line 251: "infinite-size systems". Periodic boundary conditions may represent infinite-size systems as well, or finite systems as long as one is far from the boundaries. But changing to other boundary conditions, either for finite or infinite systems, would be interesting.

17. Page 9, line 254: "The ones with odd $n$ number appear". But, in the analysis presented in this manuscript, both even and odd values of $n$ appear. What are the authors actually saying here?

18. Page 9, line 257: "reasonably robust". It seems, from the analysis presented, that as long as periodic boundary conditions are preserved, the same results would be obtained, essentially. More modes may appear, dynamics may be more complex due to this fact, but the essential features would be the same. The result discussed by the authors by doubling the size of the systems does not seem surprising.

19. Fig. 6. At this point, the "bis" notation appears for the first point, so please explain the meaning of this in the caption, or refer to the main text to understand.

20. Fig. 8: "a pseudo phase space". Why do the authors use this expression, if they have used "summary phase space" for the same axes in previous figures?

There are also some formal issues which should be addressed:

1. Please clarify units. In page 1, for instance, units are marked as $m$m. Does the difference between italics and roman font mean something? Or it should be mm? In general, units should be in roman font, so the authors should fix this in the whole text, but the additional mix of fonts in some variables is confusing.

2. In various places (page 1, for instance), products between units are marked with a dot (.). This should be either $g \cdot m^{-2}$ or (better) $g\,m^{-2}$.

3. Please fix typos

   - Spaces around parenthesis, periods, commas, spaces between words in several places in the text.
   - "A this point"
   - "Fig.06,."
   - "a an other"
   - "biommass"
   - "wit ha"
   - "pentagone"
   - "rigth"

4. Check consistency of words:

   - "Those zero mode"
   - "We need to discretization"

5. Please change words where necessary:

   - "about your system": "the" sounds better
   - "such was"

- "solution changes loses stability"
- "the branching of yet another branch"
- "which which"
- "are the indeed"

6. "3*loc*". What do the authors mean with "loc"?

7. Math variables should be in italic font.

8. Some references have incomplete bibliographic information (Meron 2015, Rietkerk 2021).

9. Fig. 5. It is hard to see that the black symbol is actually a pentagon. Please change to another, simpler shape.

10. Some decimal numbers are written with a comma, instead of a point (*e.g.* 1,13).

---

## Author Response (AR1)

**Response to the reviewers**

Lilian Vanderveken, Marina Martinez Montero and Michel Crucifix

**Response to Robbin Bastiaansen**

**MAJOR COMMENTS**

1. *As the authors point out themselves, the Rietkerk model has been studied before, and bifurcation diagrams were contructed for it before as well. Specifically, the cited paper by Zelnik et al already provides bifurcation diagrams for the Rietkerk model. Therefore, many of the figures (that is, Figures 01 to 05) are not new results (although the authors are very clear about this in the manuscript). The report on the mixed states in this model is new as far as I know. So it is a bit unfortunate that this is only covered on pages 7-8 of the paper. It would help the paper if that section could be expanded a bit, and possible be connected more to ecology or open questions in mathematical pattern formation theory.*

Thank you for your comments. As you point it out, we are well aware that a part of the results shown until page 7 are not totally new results. But as we mentionned Zelnik computes only partially the bifurcation diagram of the Rietkerk model. And we think it is still valualbe to show the full process for the linear analysis to the branch continuation in one paper. Especially because it helps to understand how the size of the domain affects the number of possible equilibria. The introduction of the zero modes in the first section is also essential to understand how the mixed state branch appears. For all those reasons, we think it is important to keep the structure of the paper as it is.

Nevertheless, we expanded the discussion about the mixed state by discussing new mixed state branches.

Lines 198-208:

The result of this routine applied to the bifurcation point of the branch n = 4 is shown on the upper panels of Fig. 6. At the transition between stability and unstability, there are three zero modes. Hence three branches of equilibria start at this transition and they are shown in Fig. 6. They begin all at the transition between the stable and unstable part of the $n = 4$ branch and they reconnect with it for higher rainfall. We call those branches $n = 4bis_1$, $n = 4bis_2$ and $n = 4bis_3$. Those three branches are close (regarding to the

mean biomass) to that of the $n = 4$ branch but if we look closely at their profile (bottom panels of Fig. 6) we see how they differ. For a given value of rainfall, the mixed state equilibria look like a modulation of the $n = 4$ equilibrium by an other wavenumber.

Equilibria branching out of zero modes in the homogeneous equilibrium tend to exhibit a single perturbation mode, see Fig. 5. By contrast, equilibria branching out of zero modes in those inhomogeneous equilibria exhibit a mixture of modes (lower panels of Fig. 6), hence the name "mixed states". The equilibria along the mixed state branch $n = 4bis_3$ are unstable, with positive eigenvalues. The branches $n = 4bis_2$ and $n = 4bis_3$ reconnect around $1.14\,\mathrm{mmd}^{-1}$.

*For instance, in Figure 08 a n=5 to n=3 transition is followed that visits one intermittent state. But what if n=6 is the starting configuration? Based on e.g. Bastiaansen & Doelman (2018) I expect that to transition to n=3 (i.e. a period doubling), but will that transient visit n=5bis first and then n=4bis before ending on n=3? In general, when there are 2N pulses and they become unstable, I expect N pulses to die off, which I would expect to go in a sort of travelling front of pulse mergings – that is, thus visiting all sort of mixed state branches. Perhaps authors can shine some light on this using their findings?*

We have repeated the experiment starting from the n=6 and n=7 branches, but due to the absence of mixed state influence on those runs we have not included additional figures. We, however, discuss the results and put them into context with the suggested literature. We discussed the transition starting from the $n = 6$ and $n = 7$ branches. For the $n = 6$, we observe a period-doubling transition directly to the $n = 3$ stable equilibrium. For the $n = 7$ branch, four bumps are removed so the the transition aapears more like a sideband destabiliziation. We also show more mixed state equilibria.

Lines 264 - 271

Now if we consider the same rainfall but we start from an other unstable branch like the $n = 6$ or $n = 7$, the dynamics is different. For $n = 6$, we observe a different type of destabilization called the period-doubling (Doelman et al. (2012), Siteur et al. (2014)). With that mechanism the system transitions to the $n = 3$ equilibrium without passing through a mixed state. And the system spends less time around the unstable equilibrium of $n = 6$ compared to the $n = 5$ case. This might be explained by the fact that the positive eigenvalue on the $n = 6$ solution is larger than the one on the $n = 5$ solution. For $n = 7$, the transition is even more rapid, as expected for an even larger positive eigenvalue, but the landing point is still, $n = 3$. In that case, we observe a destruction of four bumps similar to a sideband instability

2. *Section 3.1: here, random initial conditions are used, that are close to the bare soil state. Therefore, this procedure only samples the basin of attraction close to the bare soil state. I don't know what the purpose of that is? For instance, as rainfall decreases this is not the relevant part of the state space, but instead the regions close to the vegetation branches are.*

*In particular, in a scenario the precipitation might decrease slowly over time (e.g. conform Bastiaansen et al, 2020). So it might be more interesting to see in which basin of attraction a solution is when one of the branches destabilises. I do wonder if it is possible to completely follow a full scenario from uniform vegetation to bare soil including the pattern-to-pattern transitions and see whether they also visit the mixed states.*

The idea behind the random initial conditions experiment was to assess if the branches we have found are relevant for the dynamics. We agree that this sample only the basin of attraction close to the bare soil state and we modified the text accordingly. We decided to put this section to the appendix and take your suggestion into account by looking at the transistion from high to low rainfall started in a homogeneous equilibrium. We show that the first linear mode to destabilize according to the linear analysis is the $n = 2$, which lead to a transition to the $n = 2$ branch. The mixed state are not visited in the experiment but it was a good way to asses if the main branches that we obtained by continuation are relevant.

In the new version this new section appear as section 3.2 on page 8

3. *Lines 244-245 & lines 275-276: The importance of unstable states in ecological models has been reported on before. For instance, in Sherrat et al (2020), Morozov et al (2020), Van de Leemput et al (2015), Eigentler & Sherratt (2019), Hastings et al (2018) and Eppinga et al (2021). In the context of spatial patterns in partial differential equations this has also already been observed before, e.g. in Sherratt et al (2009). In these papers, also further references can be found. I suggest to incorporate those papers, including their terminology, to better place the paper into the context of the field.*

Thank you very much for pointing those papers to us. We included them in the discussion and use the typology and terminology to describe the mixed states.

Lines 276-291

The influence of unstable modes on dynamics has been studied in ecological models. For example Sherratt (2009) showed how spatiotemporal chaos appears in the wake of an unstable wavetrain for the complex Ginzburg-Landeau equation. In the more general context of ecology, Hastings et al.(2018) and more recently Morozov et al. (2020) proposed a classification of transient phenomenon in ecological model based on a dynamical system approach. In their definition a dynamical regime is transient if it persists for a sufficiently long time and is quasi-stable and if the transition between two regimes occurs on a much shorter timescale than the time of existence of the quasi-stable regime. According to that definition, the behavior observed in section 3.2 can be seen as a transient phenomenon. Indeed, the system spends around unstable states ($n = 5$ and $n = 4bis_3$) hundred to thousands of days and it takes only a few days to jump to the final stable equilibrium. By following the typology presented in those two papers, the transient observed here can be qualified as a 'crawl by' transient. This is due to the fact that the transient observed is linked to the existence of a saddle-type invariant set, the

mixed state. Transient dynamics have also been studied in the context of coexistence in vegetation patterns by Eigentler et al. (2019) and are characterizd by the small size of a positive eigenvalue. Long transients can also be observed in very slow front invasion dynamics (Van De Leemput et al., 2015) The mechanism behind the transient is different but the effect is the same: the system spends a long time in a region of the phase space without attractor.

**MINOR COMMENTS**

1. *Lines 16-17: Turing's article is not about vegetation patterns as is suggested in the text.*

We modified the text to avoid any confusion

Lines 16-17:

Vegetation patterns can be modelled and explained with reaction-diffusion equations. Those type of equation exhibits the existence of a homogneous stable equilibrium which is unstable to heterogeneous perturbation (Turing, 1952).

2. *Line 32: It would be good to have the model parameters in this paper explicitly as well.*

All the parametersare now in a table.

3. *Lines 42-36: In the (non-spatial) Rietkerk model there is no fold bifurcation; the 'vegetation' branch and the 'bare soil' branch coincide in a transcritical bifurcation instead. So, there you can go smoothly from one to the other, without tipping. Interestingly, the addition of spatial effects here seems to create saddle-node tipping points from patterned vegetation states to other patterned vegetation states or to the bare soil state. Hence, the text in the manuscript seems to misrepresent the findings later in the paper. I suggest to rephrase these sentences to reflect the dynamics in this Rietkerk model.*

We modified the introduction and the conclusion to make our statement about the mixed states more clear:

Line 50 - 61 :

Siteur et al. (2014) showed that a Busse balloon appears in the Klaussemeier vegetation model (Klaussemeier, 1999), where it occupies a region of the parameter space with lower rainfall than necessary to sustain a homogeneous vegetation. However, the nature of this transition through the Busse Balloon may be complex. For the non-spatial Rietkerk model, there is no such thing as a fold bifurcation. But we will see that adding a spatial component creates fold bifurcation.

The objective of the present study is to fill this gap, in the idealized context of a spatial domain of one dimension, by characterizing the intermediate states that may emerge during the transition from full vegetation to bare soil, and to examine the dynamics that underlie potential transitions between these states with a focus on a new type of equilibrium that we called "mixed state".

Specifically, we provide an in-depth analysis of the Busse balloon, demonstrate the co-existence of multiple equilibria for a given rainfall intensity, and foresee the circumstances which may trigger transitions between these different equilibria. In that sense, we propose an extension of the work by Zelnik et al. (2013) who computes partially the bifurcation diagram for Rietkerk's model. We also develop the method for finding equilibrium branches and characterize their stability. Finally, we highlight the existence of another type of equilibrium different than the regular patterns. We call them 'Mixed State' because of their shape and show how they can have an influence on the dynamics of the system.

For the non-spatial Rietkerk model, there is no fold bifurcation but as we will see later the addition of a spatial component creates a saddle-node tipping points.

Lines 326-330:

This slowing effect is well known in ecology, but to the best of our knowledge, the influence on the dynamics of the mixed states had not been previously shown for vegetation patterns. Overall, our approach allowed us to construct a bifurcation diagram that gives us valuable insights about the behavior of the system. This approach is helpful to disentangle the fate of the system in the Busse balloon and could be used to assess the existence or not of mixed states in spatially extended systems.

4. *Section 2: The stability analysis including the 'Turing' onset of patterned states has been studied before. This analysis for the here considerd Rietkerk model does appear in e.g. Siero (2020).*

We add a reference to the work of Siero at the beginning of the section

Line 66: The classical approach is to consider as in Siero et al. (2020), first, the static homogeneous equilibrium.

5. *Lines 98-101: The reasoning here is incomplete; it is not explained how it is clear that two roots are negative (which means they have negative real parts, I suppose?), and how the region in which the other one is positive can be computed. Some more detail on this should be added in the text.*

We modify the text to complete the reasonning:

Lines 102- 106:

This leads to a cubic equation for $\Omega$:

$$a_1(R,\kappa)\Omega^3 + a_2(R,\kappa)\Omega^2 + a_3(R,\kappa)\Omega + a_4(R,\kappa) = 0,$$

which we solve numerically on a physical domain defined by $\{R,\kappa\} \in \mathcal{R}$ with $R \geq 0$. This leads to three roots at each point in the domain, which we can refer to as $\Omega_1(R,\kappa)$, $\Omega_2(R,\kappa)$ and $\Omega_3(R,\kappa)$. Two of the roots, say $\Omega_{(2,3)}(R,\kappa)$, always have a real negative value, and hence correspond to exponentially decaying modes. One of the roots, $\Omega_1(R,\kappa)$, which is real in all the domain, has a positive real part in a region of the $(R,\kappa)$ plane.

6. *Lines 129-136: It reads as if a new code has been written for the continuations instead of using available continuation software. I suggest to say so explicitly in the text. Further, often people use pseudo-arclength continuation to be able to follow the folds in the branches, but the text suggests only natural continuation is used. Is this correct? In any case, I suggest to be explicit about this.*

Yes the continuation method used is the natural one. We modify the text to point that out

Lines 144-145:

A new code has been developed to implement the continuation method. This code is available online on the repository mentioned in the code availability section.

7. *Line 136: It would be good to explain why no higher n values are used: are these higher values absent, or were these not considered?*

We add a sentence to explain why we don't have more than 9 branches:

Line 144-150:

Each branch obtained by the above technique is denoted by $n$ between 1 and 9, with n the wavenumber associated with the perturbation. The periodic boundary conditions, together with a finite domain size fixed the number of zero modes present within the Turing zone , and therefore, the number of inhomogeneous branches that exist. For the periodic L=100 m domain, only nine zero modes fit within the Turing zone (see Fig. 3). Higher wavenumber modes correspond to linearly stable perturbations that exponentially decline.

8. *Lines 129-136 & lines 155-158: Since a domain with periodic boundary conditions is used, the system is translational invariant. Typically, that leads to problems in the continuation as solutions are never locally unique. How is this handled in the continuation? Further, stability of such periodic solutions could be studied using Floquet theory.*

In the continuation algorithm we set up a treshold in the incrementation at each step to avoid the Newton-Raphson method to run for an infinite amount of time.

9. *Lines 175 & 180: It is suggest by the use of the words 'may' and 'can' that something else could happen besides a bifurcation from which new branches emerge. What would that be?*

We remove the "may" and "can" because new branches always appear on a zero mode along one branch.

Line 179:

As we show next, this indicates the branching of another branch of the full-nonlinear equilibrium.

Line 184 :These zero modes act as bifurcation points from which new branches of equilibria emerge.

10. *Table 1: I do not understand the contents of this table. I suggest to add a bit more text to the caption to help interpret it.*

We remove this part of the manuscript in the new version.

11. *Figure 01: Labels for the different components are missing.*

We added labels in the figure.

12. *Figure 08: The left panel summarizes the infinite-dimensional state space via the mean biomass only. So, theoretically, not necessarily does this show that the passage is close by the mixed state equilibrium. Further, in the right panel, I find it also hard to judge how regular the transient intermittent state is (as the 'n=4bis' equilibrium solution is regularly spaced, but the simulation suggest the intermittent state is not fully regular?). So I suggest to investigate a bit further how close to the unstable equilibrium solution the transient passes in the full state space.*

The summary phase space is not defined with the mean biomass only, the maximum biomass is also used. We agree that the summary phase space is not the best option to look precisely at the distance between two solutions. But it can still help as a first insight. As suggested, in order to investigate how close to the unstable equilibrium the transient passes we show on Fig. 8 in the lower panels a snapshot of the dynamical solution at different time steps and the unstable equilibrium corresponding to $n = 4bis_3$.

We also modified the text to explain this point:

Lines 239-260:

The initial condition is an unstable equilibrium, n = 5, with a small perturbation along the direction of the first eigenvec tor. The top panels of Fig. 8 summarize the evolution of the biomass pattern over time and the corresponding trajectory in a summary phase space. This summary phase space consists of a two dimensional phase space where the two dimensions are the mean biomass and the maximum biomass. With that in mind,

we observe that the system leaves the n = 5 unstable equilibrium by first reorganising itself: mean biomass remains constant while max biomass increases. This means that one or more vegetation bumps are growing at the expense of others. After that initial reorganisation, an excursion on higher mean and max biomass takes place. It is during this out of equilibrium excursion, that one of the vegetation bumps is lost. At this point in time the vegetation profile, containing four vegetation bumps, passes close to the three mixed states originating from zero modes on then $n = 4$ branch explained on the previous section. Then, the system undergoes an other transition to the $n = 3$ equilibrium. To have a better representation of this trajectory, we also show on the upper right panel of Fig. 8 the time evolution. We see that the system quite slowly leaves the $n = 5$ unstable equilibria. The state is indeed unstable, but the associated positive eigenvalues are small, such that the dynamics around the equilibrium are slow. After abruptly departing from the $n = 5$ equilibrium, one250 biomass bump disappears, leading to a rearranged state with four bumps. On the three lower panels are represented the biomass profile for three times and their corresponding positions are also shown on the left upper panel. We see that even though the summary phase space may let us think that the system passes close to the mixed state $n = 4bis_2$, we have to keep in mind that this two dimensional space is obtained by the projection of the infinite-dimensional phase space on a two dimension summary space. On the three lower panels are represented the biomass profile for three times and their corresponding position are also shown on the left upper panel. On each of the lower panels the shape of the $n = 4bis_3$ mixed state is also shown. The shape of the dynamical solution is actually close to the $n = 4bis_3$. That equilibrium is unstable, but the system lingers in its vicinity fora considerable period of time before another bump of biomass vanishes, propelling the system towards the stable equilibrium of $n = 3$.

13. *Figure 08: The simulations are done for a value for one fixed value of the rainfall for which the n=5 branch is unstable. I wonder if this process is the same for all points on the unstable part of this branch. In particular, in e.g. Siteur et al (2014) and Doelman et al (2012), it is shown that patterned state can destabilise according to different mechanisms (sideband, period doubling, in or out of phase Hopf). Hence, it would be good to check that the reported behaviour is consistent along the unstable part of the branch.*

The reported behavior is consistent along the $n = 5$ branch in the rainfall range for which a mixed state with four bumps exist. The way the patterned state destabilises is also consistent with a sideband instability.

Lines 260-264: Switching from one state to another by losing one or more vegetation bumps is a known feature of vegetation pattern model (Bastiaansen and Doelman (2019), Bastiaansen et al. (2020)). This mechanism is called sideband instability (Doelman et al. (2012), Siteur et al. (2014)). We used the same setup with different rainfall values along the $n = 5$ branch. The fact that the system passes close to a mixed state is consistent along the $n = 5$ branch only for values of rainfall for which mixed state

equilibria exist. For rainfall lower than $\sim \mathrm{mm\,d}^{-1}$, the system jumps directly to the $n = 3$ equilibrium.

EDITORIAL

1. *Line 69: P -> B*

We fixed the typo

2. *Line 73: R -> R/d*

The inequality is now correct.

3. Section 2: I suggest to add a sentence saying you have two types of solutions to alert the reader it is not only the first one given, which the the text on line 68 now suggests.

We add a short sentecne as suggested:

Line 72: Two solutions exist depending on the value of $R$. One solution with vegetation is:

4. Equation (17): in row 2, column 1, I think the 'c' should be deleted, and in row 2, column 3 a minus sign should be added.

Thank you for pointing this. The matrix is now correct in the new version.

5. Line 172: delete either "changes" or "loses".

Done

6. Line 189: "a an other" -> "another" Done

7. Line 274: "they affect" -> "their effect"

Done

8. Figure 05: "wit ha red" -> "with a red"

Done

**Response to Reviewer #2**

1. *The manuscript provide a detailed analysis of the patterns of the model, although some results have been previously found in existing literature. I would suggest to give proper credits to previous papers.*

More references were added to properly report the existing litterature mainly mentionning the effect of unstable equilibrium in ecology and transient dynamics and the work on Daisyworld spatial extension.

Lines 275-291:

Mixed states emerge at the transition between unstable and stable states along a branch of equilibria, and have a functional form that appears as the combination of two equilibria from the main branches. We found that while these equilibria are unstable, they may still influence the system's dynamics by slowing down its evolution when it passes near them. The influence of unstable modes on dynamics has been studied in ecological models. For example Sherratt et al. (2009) showed how spatiotemporal chaos appears in the wake of an unstable wavetrain for the complex Ginzburg-Landeau equation. In the more general context of ecology, Hastings et al. (2018) and more recently Morozov et al. (2020) proposed a classification of transient phenomena in ecological model based on a dynamical system approach. In their definition, a dynamical regime is transient if it persists for a sufficiently long time (quasi-stable) and if the transition between two regimes occurs on a much shorter timescale than the time of existence of the quasi-stable regime. According to that definition, the behavior observed in section 3.2 can be seen as a transient phenomenon. Indeed, the systems spends hundred to thousands of days around unstable states(n = 5, $n = 4bis_1$ and $n = 4bis_3$) and then takes only a few days to jump to the final stable equilibrium. By following the typology presented in those two papers, the transient observed here can be qualified as a 'crawl by' transient. Indeed, the transient observed is linked to the existence of a saddle-type invariant set: the mixed state. Transient dynamics have also been studied in the context of coexistence in vegetation patterns by Eigentler and Sherratt (2019) and are characterized by the small size of a positive eigenvalue. Long transients can also be observed in very slow front invasion dynamics (Van De Leemput et al., 2015) The mechanism behind the transient is different but the effect is the same: the system spends a long time in a region of the phase space without attractor.

As we see in section 3.2, the history and the initial conditions of the system are important to see whether the mixed state will appear or not in the dynamics Other vegetation patterns models exhibit this type of sensitivity to history and initial conditions (Sherratt (2013), Adams et al. (2003) and Alberti et al. (2015)).

2. *Figure 8 (right): this is an interesting observation and it is the main result of the paper. I would suggest the authors to add more comments on this and to compare their results in terms of striped spatial patterns with those obtained for*

*energy-balance models (e.g., Adams et al., 2003, Alberti et al., 2015). This can strenghten the importance of the authors' main result in terms of inspecting the role of vegetation states into different models and contexts.*

We expanded the discussion of this part of the work. But the comparaison with the Daisy world model is difficult because in Alberti et al. 2015, the parameters are not constant in space (the solar incidence depends on the latitude) which is not the case for how the Rietkerk model was used in this work..

Lines 239-271:

The initial condition is an unstable equilibrium, n = 5, with a small perturbation along the direction of the first eigenvec tor. The top panels of Fig. 8 summarize the evolution of the biomass pattern over time and the corresponding trajectory in a summary phase space. This summary phase space consists of a two dimensional phase space where the two dimensions are the mean biomass and the maximum biomass. With that in mind, we observe that the system leaves the n = 5 unstable equilibrium by first reorganising itself: mean biomass remains constant while max biomass increases. This means that one or more vegetation bumps are growing at the expense of others. After that initial reorganisation, an excursion on higher mean and max biomass takes place. It is during this out of equilibrium excursion, that one of the vegetation bumps is lost. At this point in time the vegetation profile, containing four vegetation bumps, passes close to the three mixed states originating from zero modes on then $n = 4$ branch explained on the previous section. Then, the system undergoes an other transition to the $n = 3$ equilibrium. To have a better representation of this trajectory, we also show on the upper right panel of Fig. 8 the time evolution. We see that the system quite slowly leaves the $n = 5$ unstable equilibria. The state is indeed unstable, but the associated positive eigenvalues are small, such that the dynamics around the equilibrium are slow. After abruptly departing from the $n = 5$ equilibrium, one250 biomass bump disappears, leading to a rearranged state with four bumps. On the three lower panels are represented the biomass profile for three times and their corresponding positions are also shown on the left upper panel. We see that even though the summary phase space may let us think that the system passes close to the mixed state $n = 4bis_2$, we have to keep in mind that this two dimensional space is obtained by the projection of the infinite-dimensional phase space on a two dimension summary space. On the three lower panels are represented the biomass profile for three times and their corresponding position are also shown on the left upper panel. On each of the lower panels the shape of the $n = 4bis_3$ mixed state is also shown. The shape of the dynamical solution is actually close to the $n = 4bis_3$. That equilibrium is unstable, but the system lingers in its vicinity fora considerable period of time before another bump of biomass vanishes, propelling the system towards the stable equilibrium of $n = 3$.

Switching from one state to another by losing one or more vegetation bumps is a known feature of vegetation pattern model (Bastiaansen and Doelman (2019), Bastiaansen et al. (2020)). This mechanism is called sideband instability (Doelman et al. (2012), Siteur

et al. (2014)). We used the same setup with different rainfall values along the $n = 5$ branch. The fact that the system passes close to a mixed state is consistent along the $n = 5$ branch only for values of rainfall for which mixed state equilibria exist. For rainfall lower than $\sim \mathrm{mm\,d^{-1}}$, the system jumps directly to the $n = 3$ equilibrium. Now if we consider the same rainfall but we start from an other unstable branch like the n = 6 or n = 7, the dynamics is different. For n = 6, we observe a different type of destabilization called the period-doubling (Doelman et al. (2012), Siteur et al. (2014)). With that mechanism the system transitions to the $n = 3$ equilibrium without passing through a mixed state. And the system spends less time around the unstable equilibrium of n = 6 compared to the n = 5 case. This might be explained by the fact that the positive eigenvalue on the n = 6 solution is larger than the one on the n = 5 solution. For n = 7, the transition is even more rapid, as expected for an even larger positive eigenvalue, but the landing point is still, n = 3. In that case, we observe a destruction of four bumps similar to a sideband instability.

3.* Figures 8-9: I would suggest the authors to add more details on the possible bifurcations between different states. What about the crossing starting from a different initial equilibrium n? * We added more details about the possible transitions between states (sideband instabilities and period doublings) that could take place when considering different initial equilibrium (see previous section).

Technical comments

*Line 14: missing space between the two references.*

A space was added

*Line 24: check the style of units "mm".*

The style of the "mm" was corrected

*Line 27: check units "mm.day-1".*

The units are now correct.

*Line 30: additional/missing space before/after parenthesis.*

Fixed.

*Line 44 and Line 47: check the style of the reference.*

*Eq. (5): should be "B" instead of "P"?*

Yes, the typo is fixed

*Line 73: is there a missing "d" after k1rw?*

Yes, it is fixed in the new version

*Line 94: delete duplicated "value".*

"value" deleted

*Line 119: "to" should be "two"?*

Yes, modification done.

*Line 129: "discretization" should "discretize"?*

Yes, we modified the manuscript accordingly.

*Line 140: check units "mm.day-1".*

Units fixed

*Line 166: missing space after comma.*

Fixed.

*Line 167: "was" should be "as".*

Fixed

*Line 210: delete duplicated "which".*

"Which" is deleted.

*Line 233: also the Daisyworld models show similar features (Adams et al., 2003; Alberti et al., 2015).*

We mention now (Adams et al., 2003; Alberti et al., 2015) in the discussion section. We don't think that what we observe is exaclty the same as in (Alberti et al., 2015) because in our case we look at transient dynamics and not final states as in figure 9 of Alberti et al., 2015.

Lines 292-294: As we see in section 3.2, the history and the initial conditions of the system are important to see whether the mixed state will appear or not in the dynamics Other vegetation patterns models exhibit this type of sensitivity to history and initial conditions (Sherratt (2013), Adams et al. (2003) and Alberti et al. (2015)).

*Line 233: missing space before the second "Bastiaansen".

*Modification done.

*Line 240: this is not proper true since these findings are also observed in Daisyworld models.*

The difference with the Daisyworld models is that in our case, all the parameters of the system (Rainfall, diffusion) are homogeneous in space. In Alberti et al., 2015, there is a spatial dependency of the incoming radiation causing the change in the amplitude of daisy density.

*Figure 06: it is difficult to see the pentagone.*

We switch to a circle and increase the size of the marker.

*Figure 07: "R = 1,13" should be "R = 1,1"?*

Fixed.

**Response to Reviewer #3**

1. *Page 2, before Eq. (1). In the system of equations, all quantities should be defined (c, $g_{max}$, $k_1$, etc.).*

We added a table with all the quantities defined.

2. *Page 3, paragraph 1: "how they can have an influence on the dynamics of the system". Is it correct to say that a solution can have an "influence" on the dynamics of the system? Because a solution is an expression of the dynamics itself, it does not influence the dynamics.*

In the previous version of the text, when we were talking about "solutions", we meant static solutions, i.e., fixed points of the system. In that sense, the "solutions" have an influence on the dynamics because they can attract or repel the trajectory in their neighborhood. In the new version, we used the word equilibrium which, we hope, is much clearer.

3. *Page 3, after Eq. (10): "Fig. 01 shows these solutions as a function of rainfall R". The description in the text seems to correspond to the leftmost panel, with a vegetated solution for $R > 1$. But what do the other panels represent? Also, the second and third panels show yellow lines (solutions without vegetations) with biomass larger than zero. How this should be interpreted?*

There was a mistake with the title of each panel. Each panel respresents one variable of the system (biomass, soil water and surface water). So in the second and third panels, the yellow line can be larger than zero because it represents the soil and surface water values for the equiloibrium without vegetation.

4. *Page 4, Eq. (18). It is confusing to use letters that have been used before (c, d). Please use other names for these variables.*

We changed the letters to avoid any confusion.

Line: ?????????????????????????

This leads to a cubic equation for $\Omega$:

$$a_1(R, \kappa)\Omega^3 + a_2(R, \kappa)\Omega^2 + a_3(R, \kappa)\Omega + a_4(R, \kappa) = 0,$$

5. *Page 5, paragraph 2: "$\delta = \cos(\kappa x)$". This may be confusing, as $\delta$ has other meanings elsewhere, please change.*

We changed the notation to $\delta S(x) = \delta \cos(\kappa x)$ to be more consistent along the text.

Lines 125-126:

For each pair of $(\kappa, R)$ in the set of zero modes, we identify the corresponding homogeneous equilibrium $(\bar{S})$ and perturb it with the corresponding perturbation $\delta S \cos(\kappa x)$, and periodic boundary conditions are enforced by setting $\kappa = n\frac{2\pi}{L}$, with $n$ the wavenumber.

6. *Page 5, after Eq. (23): "This iterative procedure leads to the construction of the full branch of solutions." Could there be other nonlinear solutions, which cannot be obtained via this method? (Because they are not in the same attraction basin of the Newton algorithm, for instance.)*

In the continuation method presented here, we compute the equilibrium for one branch for a rainfall $R + \delta R$ by using the equilibrium along the same branch for a rainfall $R$ as a starting point for a Newton-Raphson method. With this approach, given a small enough $\delta R$ and being careful with possible fold bifurcations, we ensure that the algorithm converges to the next equilibrium on the same branch. To obtain other nonlinear equlibrium for the same rainfall, we use the same approach but with an other branch. So in that sense it is possible to obtain other nonlinear equilibria.

7. *Page 6, paragraph 4: "displayed on Fig.05." I do not think I understand this figure. The matrix has dimension 3, so for a given value of R and , there should be 3 eigenvalues. Why are there so many points in the lower left panel of Fig. 5?*

The matrix for the non spatial version of the model is indeed of dimension 3. But here we want to evaluate the stability of the patterned equilibrium. For this work we chose to use $N = 100$ discretization point in the spatial domain. This means that the associated Jacobian matrix should be of dimension $3N = 300$. This is why you have so many points on the left panel.

*What does the black line represent? Is it there only to mark the 0 position on the horizontal axis? This should be explained. Also, it should be explained what are the vertical and horizontal axis (imaginary and real part?).*

Indeed the black vertical line marks the 0 position. We explained the various elements on the graph and described the axes on the caption

8. *Page 6, paragraph 4: "$3\dot{1}0^{-10}$". Since eigenvalues represent frequency, do they have units? Other quantities in the system of equations do have units (R, B, etc.). *If not, how is frequency normalized? Also, please change the notation to use a centered dot ( ).*

- Eigenvalues untis are indeed frequency and so they should be $d^{-1}$. We decided to remove this part of the manuscript because the argument was not formal enough.

9. *Page 7, line 203: "we expect mixed-state modes to influence the dynamics of nearby trajectories, depending on the value of the positive eigenvalue.". This should be clarified. How can one mode "influence" the dynamics of other orbits, and what is the role of the positive eigenvalue in this.*

In the new version of the manuscript we use the words mixed state equilibrium instead of mode or solution to avoid confusion. The dynamics of the system are influenced by the presence of equilibria. That is why we use the term influence.

10. *Page 8, line 219: "the basin of attraction of the latter is narrow". Or it could be that the basin of attraction does not include regions close to the homogeneous zero state. Maybe starting from other points in phase space leads to $n = 4$, and the basin of attraction may turn out to be not small.*

YYes, you are completely right. We modified the text to clarify this point. This section was moved into the appendix to leave space for another numerical experiment with changing Rainfall parameter.

Lines 330-331: Having no trajectories on the $n = 4$ equilibrium suggests that the basin of attraction of the latter is narrow close to the homogeneous bare soil equilibrium, with few chances for a trajectory starting from random initial conditions to be in the basin of attraction of that equilibrium or, in more informal terms, to evolve such as to pass close enough to the $n = 4$ equilibrium and land on it.

11. *Page 8, line 220: $s = 4$ should be $n = 4$?*

Yes, it has been modified.

12. *Page 8, line 224: "unstable mixed-state solutions are also able to influence system transient trajectories." It is not clear that Fig. 8, which is used to explain this, really settles this. It shows that starting from an unstable mixed-state leads to an interesting dynamics. On the other hand, one should always expect that starting from an unstable solution leads to a transient trajectory, until the system is attracted to a stable solution. So it is not clear that the numerical example actually shows that mixed-state solutions "influence" trajectories. This should be better explained, and maybe rewording the statement is enough. (I am not sure that "influence" is the right word here.)*

In the experiment showed here, the starting point is not a mixed state equilibrium but the $n = 5$ equilibrium. What is interresting with this experiment is that instead of landing directly on a stable equilibrium ($n = 3$). The system spends some time around an unstable equilibrium, the mixed state $n = 4bis_3$, which is different than the initial condition. In that sense, the unstable mixed state equilibrium influences the dynamics because the system slows down around it.

13. *Page 9, table 1. Should it be "$9 \cdot 10^{-9}$" in the third column?*

We removed this part of the manuscript in the new version.

14. *Page 9, caption of table 1: "for two unstable states". It should have some reference to Fig. 08, as curves for a given value of n are not the same for different conditions.*

We removed this part of the manuscript in the new version.

15. *Page 9, line 244: "unstable modes on neighbouring dynamics is well-known in complex systems (Lucarini and Bódai, 2017)". If I understand correctly, the reference provided shows that what they call an edge state can, for instance, connect to coexistent attractors via noise induced transitions. But the authors here do not discuss the role of noise to connect two stable modes (not in Fig. 8 at least), so, again, I am not sure the authors have been clear on their claims regarding this statement.*

We modified this part of the manuscript and removed the Lucarini and Bódai, 2017 reference. We now based our discussion about the influence of unstable mixed state equilibrium on Morozov et al, 2020 and Hastings et al., 2018,.

Lines 276-291

The influence of unstable modes on dynamics has been studied in ecological models. For example Sherratt (2009) showed how spatiotemporal chaos appears in the wake of an unstable wavetrain for the complex Ginzburg-Landeau equation. In the more general context of ecology, Hastings et al.(2018) and more recently Morozov et al. (2020) proposed a classification of transient phenomenon in ecological model based on a dynamical system approach. In their definition a dynamical regime is transient if it persists for a sufficiently long time and is quasi-stable and if the transition between two regimes occurs on a much shorter timescale than the time of existence of the quasi-stable regime. According to that definition, the behavior observed in section 3.2 can be seen as a transient phenomenon. Indeed, the system spends around unstable states ($n = 5$ and $n = 4bis_3$) hundred to thousands of days and it takes only a few days to jump to the final stable equilibrium. By following the typology presented in those two papers, the transient observed here can be qualified as a 'crawl by' transient. This is due to the fact that the transient observed is linked to the existence of a saddle-type invariant set, the mixed state. Transient dynamics have also been studied in the context of coexistence in vegetation patterns by Eigentler et al. (2019) and are characterizd by the small size of a positive eigenvalue. Long transients can also be observed in very slow front invasion dynamics (Van De Leemput et al., 2015) The mechanism behind the transient is different but the effect is the same: the system spends a long time in a region of the phase space without attractor.

16. *Page 9, line 251: "infinite-size systems". Periodic boundary conditions may represent infinite-size systems as well, or finite systems as long as one is far from the boundaries. But changing to other boundary conditions, either for finite or infinite systems, would be interesting.*

The choice of periodic boundary condition discretizes the set of admissible equilibria because only a finite discrete set of wavenumbers can fit in a periodic domain. Increasing the size of the periodic domain increases the number of wavenumbers that fit the spatial domain. We modify "infinite-size" with "non perdiodic infinite-size" to be more accurate. The text is now:

Lines 288-290:

More generally, working with a periodic domain discretizes the set of admissible equilibria. Hence, the significance of mixed-states for describing the dynamics of non periodic infinite-size systems should be assessed.

17. *Page 9, line 254: "The ones with odd n number appear". But, in the analysis presented in this manuscript, both even and odd values of n appear. What are the authors actually saying here?*

We are saying that the $n$ even values branch for the L=200 m domain are the same that the ones obtained for the L=100 m domain. The $n$ odd values branches are specific to the L=200 m domain. We modified the text to be more clear about that statement.

Lines 290- 294:

As a step towards this objective, we computed the bifurcation diagram for a larger domain size of $L = 200\,\mathrm{m}$, instead of $L = 100\,\mathrm{m}$. As expected, we observed more branches in this case but, remarkably, the branches and the stability of the branches shared between the two domains are the same as for $L = 100\,\mathrm{m}$. Of course the even $n$ number branches for the $L = 200\,m$ domain are the same that the $L = 100\,\mathrm{m}$ domain. The new branches are all designated with odd $n$ number as a consequence of the extended domain.

18. *Page 9, line 257: "reasonably robust". It seems, from the analysis presented, that as long as periodic boundary conditions are preserved, the same results would be obtained, essentially. More modes may appear, dynamics may be more complex due to this fact, but the essential features would be the same. The result discussed by the authors by doubling the size of the systems does not seem surprising.*

We agree with the referee in the fact that this is not a surprising fact. We have modified the text to clarify the goal of the larger domain computation.

Lines 295-296:

These findings suggest that the existence of mixed-states and their stability properties is robust accross domain sizes, even though the way they will manifest themselves in continuous, large domains is still to be established.

19. *Fig. 6. At this point, the "bis" notation appears for the first point, so please explain the meaning of this in the caption, or refer to the main text to understand.*

We added an explanation in the caption.

20. *Fig. 8: "a pseudo phase space". Why do the authors use this expression, if they have used "summary phase space" for the same axes in previous figures?*

It was a typo in the previous version, we now use the term "summary phase space" everywhere in the text.

There are also some formal issues which should be addressed:

1. *Please clarify units. In page 1, for instance, units are marked as mm. Does the difference between italics and roman font mean something? Or it should be mm? In general, units should be in roman font, so the authors should fix this in the whole text, but the additional mix of fonts in some variables is confusing.*

We fixed the units and harmonized the notation

2. *In various places (page 1, for instance), products between units are marked with a dot (.). This should be either $textg \cdot textm^2$ or (better) $g\ m^{-2}$ .*

We followed your suggestion and modified the units accordingly.

3. *Please fix typos*

   - *Spaces around parenthesis, periods, commas, spaces between words in several places in the text.*
   - *"A this point"*
   - *"Fig.06,."*
   - *"a an other"*
   - *"biommass"*
   - *"wit ha"*
   - *"pentagone"*
   - *"rigth"*

All the typos are fixed

4. *Check consistency of words:*

   - *"Those zero mode"*
   - *"We need to discretization"*

The text was modified accordingly

5. *Please change words where necessary:*

   - *"about your system": "the" sounds better*
   - *"such was"*

The words were changed

6. *"3loc". What do the authors mean with "loc"?*

We used the 3*loc* to design a mixed state with three bumps localized. We modifiy the caption to be more explicit.

7. *Math variables should be in italic font.*

All the math variables are now in italic

8. *Some references have incomplete bibliographic information (Meron 2015, Rietkerk 2021).*

We completed the bibliography information.

9. *Fig. 5. It is hard to see that the black symbol is actually a pentagon. Please change to another, simpler shape.*

We changed to a circle.

10. *Some decimal numbers are written with a comma, instead of a point (e.g. 1,13).*

Now, all the decimal numbers are written with a dot.

LITERATURE added to the manuscript

Doelman, A., Rademacher, J. D., & van der Stelt, S. (2012). Hopf dances near the tips of Busse balloons. Discrete Contin. Dyn. Syst. Ser. S, 5(1), 61-92.

Eigentler, L., & Sherratt, J. A. (2019). Metastability as a coexistence mechanism in a model for dryland vegetation patterns. Bulletin of mathematical biology, 81(7), 2290-2322.

Eppinga, M. B., Siteur, K., Baudena, M., Reader, M. O., van't Veen, H., Anderies, J. M., & Santos, M. J. (2021). Long-term transients help explain regime shifts in consumer-renewable resource systems. Communications Earth & Environment, 2(1), 42.

Hastings, A., Abbott, K. C., Cuddington, K., Francis, T., Gellner, G., Lai, Y. C., … & Zeeman, M. L. (2018). Transient phenomena in ecology. Science, 361(6406), eaat6412.

Morozov, A., Abbott, K., Cuddington, K., Francis, T., Gellner, G., Hastings, A., … & Zeeman, M. L. (2020). Long transients in ecology: Theory and applications. Physics of Life Reviews, 32, 1-40.

Sherratt, J. A., Smith, M. J., & Rademacher, J. D. (2009). Locating the transition from periodic oscillations to spatiotemporal chaos in the wake of invasion. Proceedings of the National Academy of Sciences, 106(27), 10890-10895.

Sherratt, J. A., Liu, Q. X., & van de Koppel, J. (2021). A comparison of the "reduced losses" and "increased production" models for mussel bed dynamics. Bulletin of mathematical biology, 83(10), 99.

Siero, E. (2020). Resolving soil and surface water flux as drivers of pattern formation in Turing models of dryland vegetation: A unified approach. Physica D: Nonlinear Phenomena, 414, 132695.

van de Leemput, I. A., van Nes, E. H., & Scheffer, M. (2015). Resilience of alternative states in spatially extended ecosystems. PloS one, 10(2), e0116859.

Adams, B., Carr, J., Lenton, T. M., and White, A.: One-dimensional daisyworld: Spatial interactions and pattern formation, Journal of Theoretical Biology, 223, 505–513, 2003.

Alberti, T., Primavera, L., Vecchio, A., Lepreti, F., and Carbone, V.: Spatial interactions in a modified Daisyworld model:365 Heat diffusivity and greenhouse effects, Physical Review E - Statistical, Nonlinear, and Soft Matter Physics, 92, 1–11, 2015.